# RAPP-containing arrest peptides induce translational stalling by short circuiting the ribosomal peptidyltransferase activity

Martino Morici [1], Sara Gabrielli [2], Keigo Fujiwara [3], Helge Paternoga[1], Bertrand Beckert[1], Lars V. Bock[2], Shinobu Chiba [3] ✉ & Daniel N. Wilson [1] ✉

Arrest peptides containing RAPP (ArgAlaProPro) motifs have been discovered in both Gram-positive and Gram-negative bacteria, where they are thought to regulate expression of important protein localization machinery components. Here we determine cryo-EM structures of ribosomes stalled on RAPP arrest motifs in both *Bacillus subtilis* and *Escherichia coli*. Together with molecular dynamics simulations, our structures reveal that the RAPP motifs allow full accommodation of the A-site tRNA, but prevent the subsequent peptide bond from forming. Our data support a model where the RAP in the P-site interacts and stabilizes a single hydrogen atom on the Pro-tRNA in the A-site, thereby preventing an optimal geometry for the nucleophilic attack required for peptide bond formation to occur. This mechanism to short circuit the ribosomal peptidyltransferase activity is likely to operate for the majority of other RAPP-like arrest peptides found across diverse bacterial phylogenies.

As the nascent polypeptide chain (NC) is synthesized by the ribosome, it passes through a tunnel in the large subunit. While the ribosomal tunnel is considered a passive conduit for many NCs, in certain cases, specific interactions between the growing NC and components of the tunnel can modulate the rate of translation and even induce translational arrest[1–4]. In the past years, a number of NC-mediated translational stalling events have been shown to be part of sophisticated regulatory feedback pathways in both prokaryotes and eukaryotes[2–6]. One of the best-characterized examples is the secretion monitor (SecM) arrest peptide that regulates SecA protein expression in Gramnegative bacteria, such as *Escherichia coli*[2,7,8] (Fig. 1a). Specifically, translation arrest by SecM is thought to prevent formation of an RNA helix that normally blocks *secA* translational initiation. SecA is a motor protein that facilitates the movement of secretory proteins into and through the SecYEG protein-conducting channel[9–11]. Since the *secM* gene encodes an N-terminal signal sequence, the SecM NC is itself a substrate for SecA action. Importantly, the pulling force of SecA on the SecM NC relieves the SecM-mediated translation arrest, thereby

creating an autoregulatory feedback loop: When SecA levels are low, SecM stalling persists, leading to upregulation of *secA* expression, however, as SecA levels rise, SecM stalling is relieved, resulting in a reduction in *secA* expression[2,8] (Fig. 1a). Analogous regulatory systems have also been discovered in other bacteria, including *Bacillus subtilis* and *Vibrio alginolyticus* where the MifM and VemP arrest peptides regulate other components of the protein localization machinery, YidC2 and SecDF, respectively[12–15].

A recent bioinformatic analysis of sequenced bacterial genomes identified three additional classes of arrest peptides encoded by genes located upstream of protein localization machinery components[16] (Fig. 1b). The first two arrest peptides, termed ApcA and ApdA, were found in a subset of actinobacteria located upstream of YidC2 and SecDF, respectively, whereas the third arrest peptide, termed ApdP, was found upstream of SecDF in a subset of α-proteobacteria[16] (Fig. 1b). While ApcA and ApdA selectively arrested translation elongation on *B. subtilis* but not *E. coli* ribosomes, ApdP induced translational arrest on both *B. subtills* and *E. coli* ribosomes[16].

[1]Institute for Biochemistry and Molecular Biology, University of Hamburg, Martin-Luther-King-Platz 6, 20146 Hamburg, Germany. [2]Theoretical and Computational Biophysics Department, Max Planck Institute for Multidisciplinary Sciences, Göttingen, Germany. [3]Faculty of Life Sciences and Institute for Protein Dynamics, Kyoto Sangyo University, Kamigamo, Motoyama, Kita-ku, Kyoto 603-8555, Japan. ✉e-mail: schiba@cc.kyoto-su.ac.jp; Daniel.Wilson@chemie.uni-hamburg.de

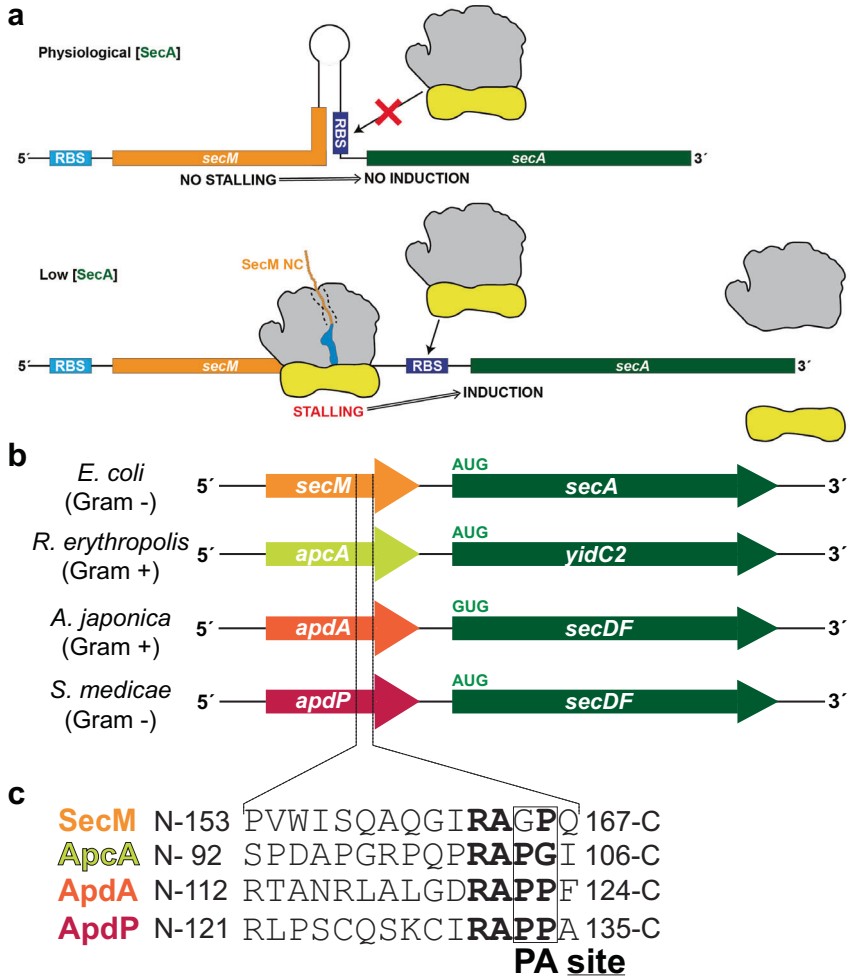

**Fig. 1 | Arrangement of bacterial regulatory operons. a** Schematic representation for the regulation of SecA by SecM. Upper panel: When SecA levels are high, the pulling force of SecA on the SecM nascent chain prevents stalling, and therefore the ribosome binding site (RBS) of the downstream *secA* gene is sequestered in a stem-loop structure, preventing SecA expression. Lower panel: When SecA levels are low, ribosomes stall during translation of SecM, leading to mRNA rearrangements that expose the RBS of the *secA* gene, leading to the expression of SecA. **b** Examples of bacterial operons containing regulatory upstream open reading frames (uORFs), including *secM-secA* in Gram-negative γ-proteobacteria, such as *Escherichia coli*, *apcA-yidC2* from Gram-positive actinomycetes, such as *Rhodococcus erythropolis*, *apdA-secDF* from Gram-positive actinomycetes, such as *Amycolatopsis japonica*, and *apdP-secDF* from Gram-negative α-proteobacteria, such as *Sinorhizobium medicae*. **c** Amino acid sequences for the SecM, ApcA, ApdA, and ApdP arrest peptides, aligned based on stalling site during translation, with A- and P-site positions indicated. Conserved residues around the stalling sites are highlighted in bold.

Biochemical analysis revealed that all three arrest peptides caused translation arrest at a conserved RAP(P/G)[16], reminiscent of the stalling at the conserved RAG/P-site reported previously for SecM[2,8,17,18] (Fig. 1c). The ApdA, ApdP and ApcA arrest peptides stall the ribosome with a peptidyl-RAP-tRNA located in the P-site and a Pro-tRNA (for ApdA and ApdP) or Gly-tRNA (for ApcA) in the A-site[16] (Fig. 1c). This indicates that, similar to SecM[2,17,19], these three arrest peptides prevent translation elongation by interfering with A-site tRNA accommodation and/or peptide bond formation[16]. A previous structural study on SecM-stalled ribosomes reported that SecM blocks the accommodation of Pro-tRNA in the A-site by inducing conformational changes within the peptidyltransferase center (PTC) that lead to an inactive state of the ribosome[20]. Moreover, the sidechain of the critically important Arg of the conserved RAG of SecM[17,18] was proposed to extend into the A-site cavity at the PTC in a manner that would sterically interfere with the placement of the Pro moiety linked to the A-site tRNA[20]. While mutagenesis studies have revealed that the Arg of the RAP motif is also critical for stalling of ApcA, ApdA, and ApdP[16], a molecular basis for the arrest mechanism used by these

arrest peptides and whether it is analogous to that reported for SecM remains to be determined.

Here we report cryo-electron microscopy (cryo-EM) structures of *B. subtilis* ApdA- and *E. coli* ApdP-stalled-ribosomal complexes (SRC) at 2.3 and 2.2 Å resolution, respectively. The structures reveal that while paths of the NCs within the ribosomal tunnel diverge for the N-terminal region, a highly conserved conformation is acquired for the C-terminal RAP/P motif at the PTC. The RAP motif of the ApdA and ApdP NCs in the P-site adopts a defined conformation that allows the PTC to acquire the induced conformation required for A-site tRNA accommodation. However, the Pro moiety on the A-site tRNA is prevented from initiating the nucleophilic attack on the P-site tRNA that would lead to peptide bond formation. Molecular dynamics (MD) simulations support a model where nucleophilic attack is prevented because a hydrogen bond between the nitrogen group of the Pro moiety in the A-site and the carbonyl-oxygen of the Ala within the RAP motif in the P-site restrains the dynamics of the Pro moiety. Although the mechanism for ApdA and ApdP is different from that reported previously for SecM[20], it is consistent with the mechanism based on a

more recent SecM structure[21]. In conclusion, our study illustrates that ApdA and ApdP arrest peptides utilize an analogous mechanism to SecM[21] to stall translation by trapping a pre-peptide bond formation (pre-attack) state of the PTC on the ribosome, and that this mechanism can operate in both Gram-positive and Gram-negative bacteria.

## Results

### Structures of *B. subtilis* ApdA-SRC and *E. coli* ApdP-SRC

Since ApdA was shown to stall efficiently on *B. subtilis*, but not *E. coli*, 70S ribosomes[16], we employed a *B. subtilis* cell-free in vitro translation system to generate ApdA-stalled-ribosomal complexes (ApdA-SRC), as used previously to generate *B. subtilis* MifM-SRCs[22]. In contrast to the MifM-SRC[22], the ApdA-SRC was purified using an N-terminal FLAG-tag exposed at the exit tunnel and was formed with full-length, rather than truncated, mRNA (see Methods). The ApdA-SRC was applied to cryo-

grids and analyzed using single-particle cryo-EM. A total of 9930 micrographs were collected on a Titan Krios transmission electron microscope (TEM) equipped with a K3 direct electron detector (DED), which yielded 334,479 ribosomal particles after 2D classification (Supplementary Fig. 1). Focused 3D classification revealed two major subpopulations of 70S ribosomes, one bearing A- and P-site tRNAs (43%; 142,978 particles) and one with only P-site tRNA (46%; 152,257 particles), collectively representing a total of 89% of the initial ribosomal particles (Supplementary Fig. 1). The 70S ribosome with A- and P-site tRNAs was further refined, yielding a cryo-EM map of the ApdA-stalled-ribosomal complex (ApdA-SRC) with an average resolution of 2.3 Å (Fig. 2a, Supplementary Figs. 1, 2, and Table 1). In the ApdA-SRC, the ApdA nascent polypeptide chain (NC) was well-resolved near the PTC (Supplementary Fig. 2 and Supplementary Movie 1), such that nine amino acids, including sidechains, could be modeled unambiguously,

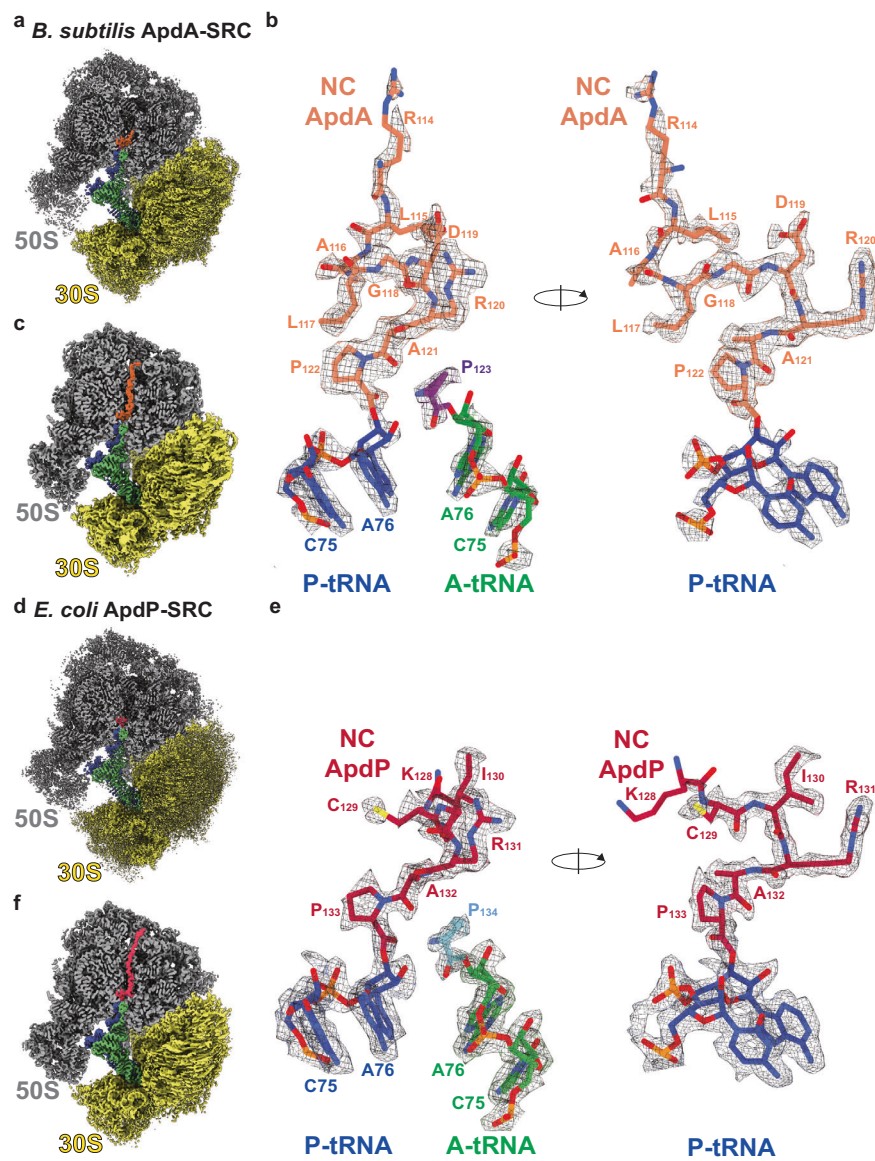

**Fig. 2 | Cryo-EM structures of ApdA- and ApdP-SRCs. a** Cryo-EM map of the post-processed *B. subtilis* ApdA-SRC with a transverse section of the 50S (gray) to reveal density for the nascent chain (orange), P-tRNA (blue), A-tRNA (green); 30S (yellow). **b** Two views showing the cryo-EM map density for A- and P-site tRNAs of the post-processed *B. subtilis* ApdA-SRC. The P-site tRNA (blue) bears the ApdA nascent chain (orange), whereas the A-site tRNA (green) carries proline (purple). **c** as (**a**), but cryo-EM map of 3D-refined *B. subtilis* ApdA-SRC. **d** Cryo-EM map of the post-

processed *E. coli* ApdP-SRC with a transverse section of the 50S (gray) to reveal density for the nascent chain (red), P-tRNA (blue), A-tRNA (green); 30S (yellow). **e** Two views showing cryo-EM map density for A- and P-site tRNAs of the post-processed *E. coli* ApdP-SRC. The P-site tRNA (blue) bears the ApdP nascent chain (red), whereas the A-site tRNA (green) carries proline (cyan). **f** as (**d**), but cryo-EM map of 3D-refined *E. coli* ApdP-SRC.

**Table 1 | Cryo-EM data collection, refinement and validation statistics**

| Complex | *B. subtilis* ApdA-SRC | *E. coli* ApdP-SRC |
|---|---|---|
| EMDB ID | 18332 | 18320 |
| PDB ID | 8QCQ | 8QBT |
| **Data collection and processing** | | |
| Magnification (×) | 105,000 | 105,000 |
| Acceleration voltage (kV) | 300 | 300 |
| Electron fluence (e⁻/Å²) | 42 | 42 |
| Defocus range (μm) | −0.6 to −1.8 | −0.6 to −1.8 |
| Pixel size (Å) | 0.82 | 0.82 |
| Symmetry imposed | C1 | C1 |
| Initial particles | 334,479 | 263,503 |
| Final particles | 142,978 | 205,838 |
| Average resolution (Å) (FSC threshold 0.143) | 2.3 | 2.2 |
| **Model composition** | | |
| Initial model used (PDB code) | 6HA1 | 5JTE |
| Atoms | 136,962 | 141,132 |
| Protein residues | 4822 | 4848 |
| RNA bases | 4564 | 4738 |
| **Refinement** | | |
| Map CC around atoms | 0.76 | 0.69 |
| Map CC whole unit cell | 0.74 | 0.67 |
| Map sharpening B factor (Å²) | −45.24 | −46.75 |
| **R.M.S. deviations** | | |
| Bond lengths (Å) | 0.009 | 0.010 |
| Bond angles (°) | 1.717 | 1.727 |
| **Validation** | | |
| MolProbity score | 1.08 | 1.16 |
| Clash score | 0.78 | 0.91 |
| Poor rotamers (%) | 0.94 | 0.78 |
| **Ramachandran statistics** | | |
| Favored (%) | 95.22 | 94.39 |
| Allowed (%) | 4.54 | 5.42 |
| Outlier (%) | 0.23 | 0.19 |
| Ramachandran Z-score | −2.30 | −2.04 |

covering the C-terminal conserved $_{120}$RAP$_{122}$ motif that is directly linked to the CCA-end of the P-site tRNA (Fig. 2b and Supplementary Fig. 2). In addition, the high quality of the cryo-EM map density allowed the Pro$_{123}$ moiety attached to the CCA-end of the A-site tRNA to be unambiguously identified and modeled (Fig. 2b). Although additional density for the ApdA NC could also be observed throughout the entirety of the ribosomal exit tunnel in some pre-processed maps (Fig. 2c), it was poorly resolved, indicating flexibility, and precluded a molecular model to be generated for these regions.

Unlike ApdA, ApdP was shown to stall efficiently on both *B. subtilis* and *E. coli* 70S ribosomes[16]; therefore, we generated ApdP-stalled-ribosomal complexes (ApdP-SRC), as used previously to generate *E. coli* SecM-SRC[23] and VemP-SRC[24]. As for ApdA-SRC, the ApdP-SRC was purified using an N-terminal FLAG-tag and full-length mRNA, and analyzed by single-particle cryo-EM using a Titan Krios TEM equipped with a K3 DED (see Methods). A total of 4921 micrographs were collected, which yielded 263,503 ribosomal particles after 2D classification (Supplementary Fig. 3 and Table 1). Focused 3D classification revealed one major subpopulation of 70S ribosomes bearing A- and P-site tRNAs (78%; 205,838 particles), as well as one minor population with P-site tRNA only (7%; 17,657 particles), collectively representing a

total of 85% of the initial ribosomal particles (Supplementary Fig. 3). The 70S ribosome with A- and P-site tRNAs were further refined, yielding a cryo-EM map of the ApdP-stalled-ribosomal complex (ApdP-SRC) with an average resolution of 2.2 Å (Fig. 2d and Supplementary Fig. 4). In the ApdP-SRC, the ApdP NC was well-resolved near the PTC (Supplementary Fig. 4 and Supplementary Movie 2), such that six amino acids, including sidechains, could be modeled, including the C-terminally conserved $_{131}$RAP$_{133}$ motif that is directly linked to the CCA-end of the P-site tRNA (Fig. 2e and Supplementary Fig. 4). As for the ApdA-SRC, the cryo-EM map density of the ApdP-SRC allowed the unambiguous modeling of Pro$_{134}$ attached to the CCA-end of the A-site tRNA (Fig. 2e), whereas the additional density for the ApdP NC observed in the deeper regions of the ribosomal exit tunnel in some pre-processed maps was poorly resolved (Fig. 2f), precluding a molecular model to be built.

We also refined the ApdA- and ApdP-ribosome populations lacking A-site tRNA, yielding cryo-EM maps of the ApdA-(ΔA-tRNA)-SRC and ApdP-(ΔA-tRNA)-SRC with average resolutions of 2.3 and 2.9 Å, respectively (Supplementary Fig. 5). Although the ribosome and P-site tRNA were well-resolved, in both cases the density for the NC, even directly at the PTC, was poorly defined, preventing molecular models to be generated. Alignment with the ApdA- and ApdP-SRCs indicated that some density is observed for the RAP motif attached to the P-site tRNA and that the overall conformation appears to be similar to that observed for ApdA/ApdP from the ApdA/ApdP-SRCs, but with higher flexibility (Supplementary Fig. 5). Thus, we conclude that the presence of the A-site tRNA in the ApdA- and ApdP-SRC stabilizes the conformation of the NC on the peptidyl-tRNA in the P-site. Collectively, the cryo-EM structures of the ApdA- and ApdP-SRC revealed that the peptidyl-tRNA is present in the P-site, consistent with previous biochemical analysis[16], supporting the suggestion that the ApdA and ApdP arrest peptides interfere with peptide bond formation between the peptidyl-RAP-tRNA in the P-site and the incoming A-site Pro-tRNA[16].

## Interaction of ApdA and ApdP at the ribosomal PTC

Superimposition of the ApdA- and ApdP-SRCs reveals that the RAP motif attached to the P-site tRNA is positioned identically (within the limits of the resolution) between the two structures (Fig. 3a). Similarly, the interactions of the RAP motif of ApdP with the 23S rRNA nucleotides located at the PTC of the *E. coli* ribosome (Fig. 3b) are indistinguishable from that observed between the RAP motif of the ApdA and the *B. subtilis* ribosome (Fig. 3c). This finding is consistent with the high sequence and structural conservation of these nucleotides (Fig. 3b, c). With the exception of a potential hydrogen bond between the backbone nitrogen of Ala132 of ApdP (Ala121 in ApdA) and U2506 (BsU2535), the RAP motif interactions with the surrounding 23 S rRNA involve exclusively the Arg131 of ApdP (Arg120 in ApdA) (Fig. 3b, c). Specifically, the Arg of the RAP motif inserts into a pocket formed by 23S rRNA nucleotides of the PTC, where the sidechain stacks upon Ψ2504 (BsU2533) (Fig. 3b–d) and can form five direct hydrogen bonds to rRNA, three with the nucleobases of Ψ2504 (BsU2533) and G2061 (BsG2090) as well as two with the phosphate-oxygen backbone of G2505 (BsG2534) (Fig. 3b, c). In addition, we observe density for two water molecules that mediate interactions between the Arg of the RAP motif and 23S rRNA nucleotides G2505 (BsG2534), m²A2503 (Bs m²A2532), and G2061 (BsG2090) (Fig. 3b, c). The importance of these interactions is supported by the observation that mutation of Arg120 to Ala in ApdA abolishes stalling in *B. subtilis*, and mutation of Arg131 to Ala in ApdP abolishes stalling in both *B. subtilis* and *E. coli*[16]. By contrast, mutation of Ala121 in ApdA, and Ala132 in ApdP, to Ser had a less dramatic effect on the stalling efficiency[16], consistent with the backbone interaction of the Ala of the RAP motif observed in the respective structures (Fig. 3b, c).

Given the ability of the RAPP motif to stall translation in both Gram-positive and Gram-negative bacteria, we also tested whether the

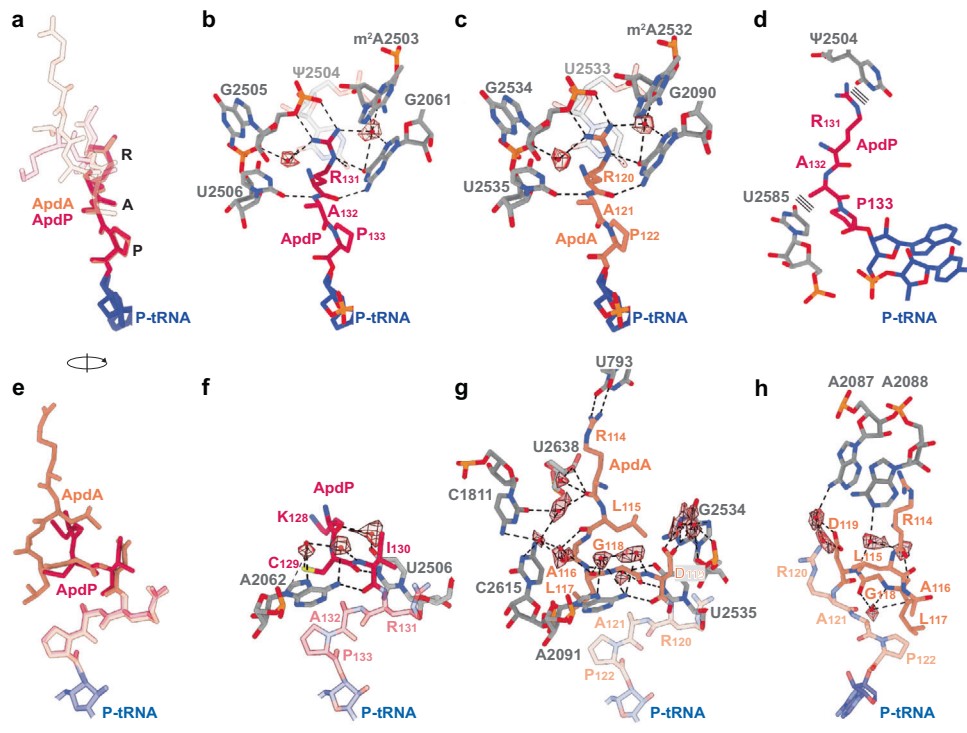

**Fig. 3 | Interaction of ApdA and ApdP NCs within the ribosomal tunnel.**
**a** Overlay (aligned on the basis of the 23S rRNA) of the molecular models for ApdA (orange), ApdP (red), and P-tRNA (blue) with a focus on the RAP motif.
**b**, **c** Interactions of RAP motif of the **b** ApdP (red) and **c** ApdA (orange) nascent chains with the exit tunnel nucleotides (gray) of the **b** *E. coli* and **c** *B. subtilis* 23S rRNA, respectively. Potential hydrogen bonds are shown as dashed lines and water molecules as red spheres (with meshed density). **d** Rotated view of (**b**) showing the

stacking interactions (depicted as three parallel lines) between the conserved RAP motif of ApdP (red) and 23S rRNA nucleotides (gray). In (**b**) and (**d**), ψ indicates the presence of pseudouridine at position 2504 in *E. coli*. **e** Rotated view of (**a**) with a focus on residues N-terminal to the RAP motif. **f**–**h** Interactions (**f**) ApdP (red) and (**g**, **h**) ApdA (orange) nascent chains with the exit tunnel nucleotides (gray) of the **f** *E. coli* and **g**, **h** *B. subtilis* 23S rRNA, respectively. Potential hydrogen bonds are shown as dashed lines and water molecules as red spheres (with meshed density).

RAPP motif could induce translational stalling in a eukaryotic system. To do this, we introduced the C-terminal soluble domains of wildtype ApdA and ApdP, as well as their variants where the RAPP motif was mutated to AAPP, into a GFP-LacZ reporter and monitored for presence of peptidyl-tRNA and full-length protein after incubation in a rabbit reticulocyte in vitro translation system (Supplementary Fig. 6). As a positive control, we employed the XBP1u arrest peptide, where we observed stalling, as expected[25], indicated by the accumulation of peptidyl-tRNA that is resolved upon addition of RNase (Supplementary Fig. 6a). By contrast, we observed no accumulation of peptidyl-tRNA caused by elongation arrest using the ApdA or ApdP sequences, suggesting that the RxPP motif does not stall eukaryotic translation efficiently. One explanation is that in ApdA and ApdP, the critically important Arg of the RxPP motif stacks upon Ψ2504 (BsU2533) (Fig. 3b–d), however, in eukaryotic ribosomes, the equivalent rRNA nucleotide, U4412, adopts a different conformation that precludes this stacking interaction (Supplementary Fig. 6b).

In contrast to the RAPP motif, the conformation of the modeled residues located N-terminally is distinct when comparing ApdA and ApdP (Fig. 3e), which is not unexpected given the lack of sequence homology in this region (Fig. 1c). The three N-terminal residues (Lys128-Ile130) observed for ApdP form a network of direct and water-mediated interactions with A2062 and U2506 of the 23S rRNA (Fig. 3f). Similar interactions are observed for the corresponding region of ApdA (Leu117-Asp119) with direct and water-mediated interactions observed to A2091 (EcA2062) and G2534 (EcU2505) (Fig. 3g). Unlike for ApdP, we observe an additional three residues (Arg114–Ala116) of ApdA that establish direct and water-mediated interactions with U793

(EcΨ746), C1811 (EcU1782), C2615 (EcU2586), U2638 (EcU2609) (Fig. 3g), and A2087/A2088 (EcA2058/A2059) (Fig. 3h). While single point mutation of these residues to alanine had no influence on translational stalling[16], frameshifted constructs indicate that residues within this N-terminal region, namely, residues 125–130 for ApdP and 108–119 for ApdA, do contribute to the efficiency of translational stalling[16]. These observations suggest that while individual interactions are unlikely to be important, collectively, this N-terminal region also plays a role in facilitating translational stalling of both ApdA and ApdP.

**The species-specificity determining region of ApdA and ApdP**
Since previous studies demonstrated that ApdP induces efficient translational stalling in both *E. coli* and *B. subtilis*, whereas ApdA exhibits efficient stalling only in *B. subtilis*[16], we generated a series of ApdA-ApdP chimeric constructs (Fig. 4a) and monitored whether stalling is observed using *B. subtilis* and *E. coli* in vitro translation systems (Fig. 4b, c). As expected, all ApdA-ApdP chimeric constructs induced efficient stalling on *B. subtilis* ribosomes (Fig. 4b, lanes 1–20), whereas different efficiencies of translational stalling were observed on *E. coli* ribosomes (Fig. 4c, lanes 1–20). Compared to the wildtype ApdP sequence (Fig. 4c, lanes 1–2), the substitution of ApdP residues N-terminal to the RAPP motif with the corresponding ApdA sequence led to reduced stalling, as observed by the reduction of peptidyl-tRNA and the appearance of full-length (FL) protein (Fig. 4c, lanes 3–8). Loss of ApdP stalling on *E. coli* ribosomes was observed with the ApdP125-A115 construct (Fig. 4c, lanes 7–8) when only five residues of ApdP ($_{126}$QSKCI$_{130}$) were replaced with the corresponding residues from ApdA ($_{115}$LALGD$_{119}$) (Fig. 4a). Conversely, compared to the wildtype

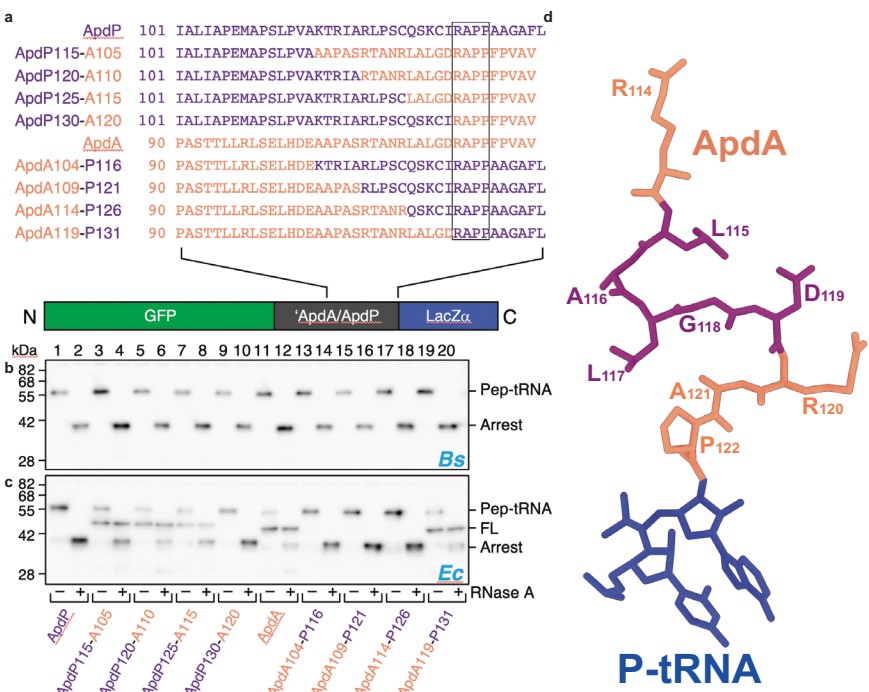

**Fig. 4 | Species-specific stalling of chimeric ApdA-ApdP constructs. a** Sequences of different chimeras between ApdA (orange) and ApdP (purple) (with the common RAPP motif boxed) cloned in the GFP-LacZα construct shown at the bottom. **b, c** Western blot against GFP showing the outcome of the stalling assay in **b** *B. subtilis* and **c** *E. coli* in vitro translation system for the chimeras listed in (**a**); each reaction was loaded before (−) and after (+) RNase A treatment. Bands corresponding to peptidyl-tRNA (Pep-tRNA), full-length peptide (FL), and truncated peptide arising due to the stalling (Arrest) are indicated. Experiments were performed in two independent experiments with similar results. Source data are provided as a Source Data file. **d** Structure of ApdA (orange) with residues colored purple that enhanced stalling on *E. coli* ribosomes when substituted with the corresponding ApdP residues.

ApdA sequence (Fig. 4c, lanes 11–12), substitution of ApdA residues N-terminal to the RAPP motif with the corresponding ApdP sequence led to increased stalling, as observed by the increase in peptidyl-tRNA and the loss of full-length (FL) protein (Fig. 4c, lanes 13–18). Here, the gain of ApdA stalling on *E. coli* ribosomes was observed with the ApdA114-P126 construct (Fig. 4c, lanes 17–18) when only five residues of ApdA ($_{115}$LALGD$_{119}$) were replaced with the corresponding residues from ApdP ($_{126}$QSKCI$_{130}$) (Fig. 4a). Collectively, this suggested that one or more of the five residues located within the region directly N-terminal to the RAPP motif (Fig. 4d) play(s) a critical role in the species-specificity of ApdA translational stalling.

A comparison of the 23S rRNA nucleotides that surround and interact with these five ApdA residues ($_{115}$LALGD$_{119}$) in the ApdA-SRC with the equivalent 23S rRNA nucleotides in the *E. coli* ribosome revealed no discernable conformational differences that would provide an explanation for the species-specific stalling (Supplementary Fig. 7a, b). We note that while there is one sequence difference between *B. subtilis* and *E. coli* ribosomes within this region, namely, the equivalent nucleotide to C2615 in the *B. subtilis* ribosome is U2586 in the *E. coli* ribosome, we do not believe that it is responsible for the species-specificity of ApdA since the 23S rRNA of *A. japonica*, where ApdA was discovered[16], contains a U rather than a C at the equivalent position. Collectively, this suggests that more distal regions must cooperate with the proximal five residues to influence species-specificity. Indeed, the ability of MifM to stall on *B. subtilis* ribosomes, but not *E. coli* ribosomes, was shown to arise due to sequence differences between ribosomal protein uL22, with one residue (Met90) of uL22 playing a critical role[22]. Although the ApdA and ApdP NCs were not resolved sufficiently in the distal regions of the tunnel to generate a molecular model, a comparison of the cryo-EM densities for the ApdA and ApdP NCs indicates that their paths may deviate in proximity to the loops of ribosomal proteins uL4, uL22, and uL23 (Supplementary

Fig. 7c, d). Therefore, to assess whether these ribosomal proteins contribute to the stalling efficiency of ApdA and ApdP in *B. subtilis*, we monitored stalling in vivo using wildtype *B. subtilis* as well as *B. subtilis* with alterations in ribosomal proteins uL4, uL22, and uL23 (Supplementary Fig. 7e–h). As a control, we employed a reporter with the MifM stalling sequence, where stalling was observed in the wildtype *B. subtilis* strain as well as a *B. subtilis* strain with a loop deletion (Δ65-69) in uL23, but was impaired in *B. subtilis* strains with loop deletions in uL4 (Δ66–70) or uL22 (Δ86–90), or when the *B. subtilis* uL22 loop was substituted with the *E. coli* sequence (Ec-loop), as reported previously[22]. In contrast to MifM, stalling due to ApdA and ApdP remained completely unaffected in all strains regardless of deletions or substitutions in the loops of uL4, uL22, and uL23 (Supplementary Fig. 7e–h). This suggests that, unlike MifM, the interaction of ApdA and ApdP with the loops of uL4, uL22, and uL23 does not appear to be critical for their mechanism of translational stalling or species-specificity. However, we cannot rule out that interactions of other regions of uL4, uL22, and uL23 with the ApdA NC may contribute to the species-specificity of stalling of ApdA.

**Interaction of ApdA and ApdP NCs with the A-site proline**
To understand how the ApdA and ApdP arrest peptides interfere with peptide bond formation, we compared the conformation of the A- and P-site tRNAs and surrounding 23S rRNA at the PTCs of the ApdA- and ApdP-SRC with the previous structures of pre-attack state ribosomes at 2.5–2.6 Å[26,27] (Fig. 5a–f and Supplementary Figs. 8, 9). We note that the conformation of the 23S rRNA at the PTC of the pre-attack state appears to be identical (within the limits of the resolution) with the 23S rRNA of the ApdA- and ApdP-SRC (Supplementary Fig. 9), indicating that the PTC of the ApdA- and ApdP-SRC have adopted the induced state that is necessary for full accommodation of the A-site tRNA[26,28]. During peptide bond formation, the α-amino group of the aminoacyl

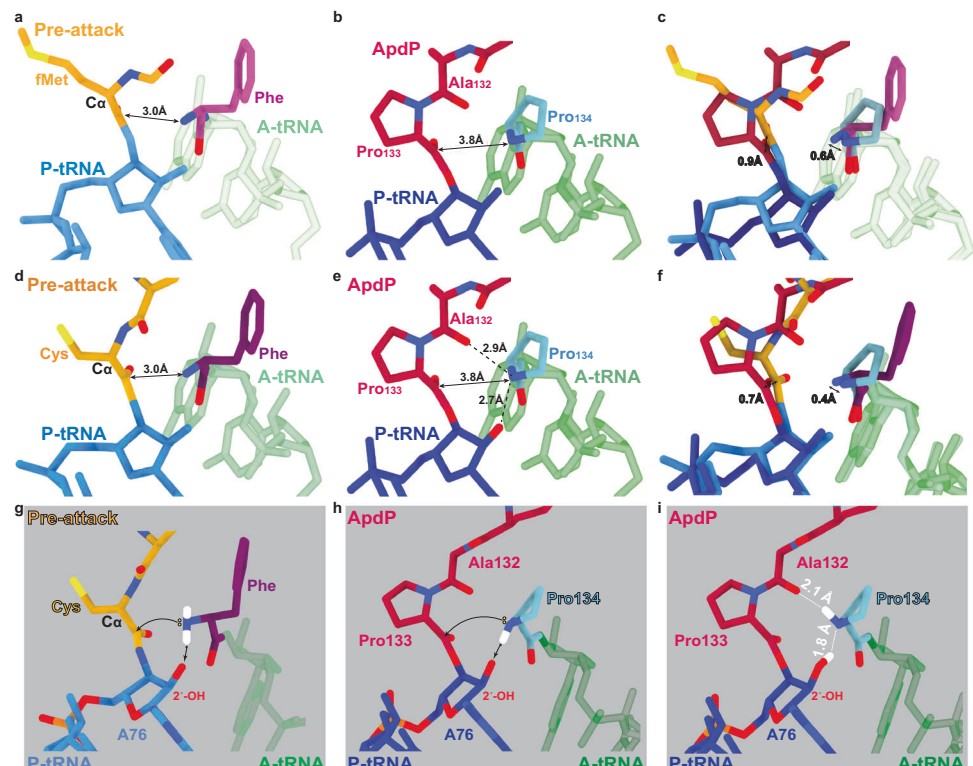

**Fig. 5 | ApdA/ApdP stabilize the pre-attack state at the PTC. a** View of the PTC of a pre-attack state (PDB ID 1VY4)[26], showing a fMet-NH-tRNA (gold/blue) at the P-site and a phenylyl-NH-tRNA (purple/green) at the A-site. The distance (3.0 Å) between the attacking amine of the A-tRNA and the carbonyl-carbon of the P-tRNA is arrowed. **b** Same view as (**a**), but for the ApdP-SRC with ApdP-tRNA (red/dark blue) in the P-site and Pro-tRNA (cyan/green) in the A-site. **c** Overlay of (**a**, **b**) (aligned on the basis of the 23S rRNA) highlighting the difference in the distance between the attacking nitrogen groups of A-site aminoacyl moiety and the carbonyl carbons at the P-site. **d** View of the PTC of a pre-attack state (PDB ID 8CVK)[27], showing a tripeptidyl-NH-tRNA (gold/blue) at the P-site and a phenyl-NH-tRNA (purple/green) at the A-site. The distance (3.0 Å) between the attacking nitrogen of the A-tRNA and the carbonyl-carbon of the P-tRNA is arrowed. **e** Same view as (**d**), but for the ApdP-SRC with ApdP-tRNA (red/dark blue) in the P-site and Pro-tRNA (cyan/green) in the

A-site. **f** Overlay of (**d**, **e**) (aligned on the basis of the 23S rRNA) highlighting the difference in the distance between the attacking nitrogen groups at the A-site and the carbonyl carbons at the P-site. **g** Schematic view of the PTC of a pre-attack from (**d**), but with hydrogen atoms (white) modeled in silico for the α-amino group of the Phe moiety in the A-site. The yellow spheres indicate the lone pair electrons that make the nucleophilic attack (arrowed) on the carbonyl-carbon of the fMet moiety on the P-site tRNA. **h** Same schematic as (**b**), but with the hydrogen atom (white) modeled in silico towards the 2'OH of A76 of the P-site tRNA, which would allow a nucleophilic attack (arrowed) on the carbonyl-carbon of the Pro133 moiety attached to the P-site tRNA. **i** Same schematic as (**h**) but with the hydrogen atom (white) modeled toward the carbonyl of Ala132, a conformation that would prohibit any nucleophilic attack.

moiety of the A-site tRNA makes a nucleophilic attack on the carbonyl-carbon of the peptidyl-tRNA in the P-site[29–31]. In the pre-attack state structures, the α-amino group of Phe-tRNA in the A-site is located 3.0 Å from the carbonyl-carbon of the fMet-tRNA in the P-site; however, no peptide bond formation can occur because the fMet moiety is linked to the P-site tRNA via an amide, rather than an ester, linkage[26,27] (Fig. 5a, c). In the ApdA- and ApdP-SRC, the α-amino group of Pro-tRNA in the A-site is located 3.7 and 3.8 Å, respectively, from the carbonyl-carbon of the peptidyl-RAP-tRNA in the P-site (Fig. 5b and Supplementary Fig. 8), however, peptide bond formation has not occurred, despite the existence of an ester linkage between the NC and the P-site tRNA. Superimposition of the pre-attack states with the ApdA- or ApdP-SRC suggests that the enlarged distance between the α-amino group and the carbonyl-carbon results from a slightly shifted position in both the A- and P-site moieties, but not in the A- and P-site tRNAs themselves (Fig. 5c, f and Supplementary Fig. 8), however, exact quantification is not possible, even at this resolution. Collectively, these observations indicate that the translation of ApdA and ApdP becomes stalled even though the PTC is in the canonical induced state, suggesting that the process of peptide bond formation itself is directly affected.

In the current models for peptide bond formation[26,32–34], a proton is extracted from the attacking α-amino group by the 2'O of the A76 of

the P-site tRNA, which increases the nucleophilicity of the α-amino group and thereby facilitates the nucleophilic attack (Fig. 5g). For stalling of ApdA and ApdP, proline (Pro134 for ApdP and Pro123 for ApdA) in the A site is required. In contrast to all other proteinogenic amino acids, Pro does not have a primary amino group, but rather is a secondary amine, such that the nitrogen carries only one hydrogen in the uncharged state (Fig. 5h). The rate of peptidyl transfer when the Pro-tRNA is in the A-site is low[35] and it has been suggested that the elevated pKa of Pro might contribute to the reduced cumulative rate of steps leading to peptidyl transfer after GTP hydrolysis on EF-Tu[36]. However, the exact protonation state of Pro prior to peptidyl transfer is unclear. While the protonation state cannot be directly inferred from our cryo-EM structures because hydrogens are not resolved, the position of hydrogens can be predicted based on hydrogen-bonding distances. For example, in the ApdA- and ApdP-SRC structures, the N of the A-site Pro is in close proximity to both the carbonyl-oxygen of Ala132 of ApdP (2.7–2.9 Å) as well as the 2'O of the P-site tRNA A76 ribose (2.6–2.7 Å) (Fig. 5e and Supplementary Fig. 8). In the case of an uncharged Pro, it is only possible to form these two strong hydrogen bond interactions when the single hydrogen of the nitrogen of Pro acts as a donor with the carbonyl-oxygen of Ala132, and the nitrogen as an acceptor for the hydrogen from the 2'O of A76 (Fig. 5i)—a scenario that is not compatible with peptide bond formation because the 2'O cannot

extract a proton from the Pro. An alternative scenario is that the Pro is protonated, i.e., presenting two hydrogens, and therefore could act as a donor for both the carbonyl-oxygen of Ala132 as well as the 2'O of A76 (Supplementary Fig. 10). However, also in this scenario where a proton could be extracted by the ribose 2'O, the hydrogen bond remaining with the carbonyl-oxygen of Ala132 maintains a geometry that is incompatible with nucleophilic attack by the lone pair electrons (Supplementary Fig. 10). Taken together, this suggests that translation of ApdA and ApdP becomes stalled because the nucleophilic attack of the A-site Pro moiety on the carbonyl-carbon of the P-site peptidyl-tRNA is blocked, and thus peptide bond formation cannot ensue.

## MD simulations of the ApdP-SRC

To test our predictions, we initiated MD simulations from the ApdP-SRC cryo-EM model with different Pro134 protonation states and compared the resulting ensemble with the cryo-EM model. Three protonation states were simulated, (i) ↖Pro134, where the N-H of the uncharged Pro is oriented towards the carbonyl-O (Ala132), (ii) ↗Pro134, where the N-H points towards the ribose O2' (A76), and (iii) Pro⁺134, where the charged state of Pro has two N-H forming interactions with both the carbonyl-O (Ala132) and the ribose O2' (A76). For the correct scenario, the structural ensemble generated by the MD simulations is expected to remain close to the cryo-EM model, while for the other scenarios, the protonation state could lead to conformational changes and, therefore, larger structural deviations. To quantify the deviation from the cryo-EM model, we calculated the root mean square deviation (rmsd) of the P-site tRNA A76 and of the two C-terminal amino acids of ApdP (Pro133 and Pro134) from their cryo-EM conformation (Fig. 6a). While the distribution of rmsd values of Pro133 is similar for all three scenarios, a clear shift towards larger rmsd values was observed for Pro134 and A76 in the Pro⁺134 state.

These results suggested that a protonated Pro134 causes Pro133 and A76 to undergo conformational changes that are incompatible with the cryo-EM model, rendering this scenario highly unlikely. While the distribution of rmsd values of Pro133, Pro134, and A76 are similar for the two uncharged protonation states of Pro134, the N(Pro134)-C(Pro133) distance between the α-amino group of Pro134 and the carbonyl-carbon of Pro133 sampled by the ↗Pro134 protonation state largely deviates from the one observed in the cryo-EM structure (Fig. 6b). Hence, in agreement with our predictions, the structural ensemble that is closest to the cryo-EM model is the one obtained for the ↖Pro134 protonation state (Fig. 6c).

For a peptide bond to form, two conditions must be met: Firstly, a short distance between the N and C atoms involved in the peptide bond must arise, presumably shifting towards the distance (3.0 Å) observed in the pre-attack structures[26,27], and, secondly, the α-amino N must lose one hydrogen (proton), which, as mentioned before, is mediated by the 2'O of A76[26,32-34]. We propose that the reason for ApdP stalling is that even if the N-C distance is small, the A-site amino acid and the P-site peptide are hindered from reaching conformations that allow the proton transfer from the N-H (Pro134) to the 2'O of A76. It has been shown that mutations in the RAPP motif of ApdP can alleviate stalling[16], presumably because the dynamics for non-stalling mutants change, enabling productive conformations to be visited that allow peptide bond formation. To test this, we performed additional MD simulations with non-stalling ApdP variants R131A, A132S, P133A, and P134A. As a negative control, we performed MD of the K128A variant that did not affect the ApdP stalling efficiency[16]. To check if productive conformations were reached for the ApdP variants, we identified the conformations that satisfy all the conditions required for peptide bond formation. Firstly, to check for the proximity requirement, we counted how often conformations with N(Pro134)-C(Pro133) distance lower than 3.8 Å were sampled. The threshold of 3.8 Å was chosen as it is the

distance observed in the cryo-EM structure. As shown in Fig. 6d (magenta bars), conformations meeting the first requirement are sampled in all variants with frequencies in the same order of magnitude. However, marked differences between stalling and non-stalling variants arise when, in addition to the proximity condition, also the second condition for peptide bond formation, i.e., the N-H(Pro134) −2'O(A76) hydrogen bond, is met (Fig. 6d, blue bars). The differences are even more pronounced when considering also the 2'OH(A76) −2'O(A2451) hydrogen bond (Fig. 6d, yellow bars), which constitutes the second step in the proton wire mechanism[26]. Interestingly, while in the wildtype and control, the three conditions are rarely met simultaneously, the non-stalling variants meet all requirements 100- to 1000-fold more often. This confirms our prediction that, despite reaching short N(Pro134)-C(Pro133) distances, stalling of the wildtype and control takes place due to the inability of the ApdP peptide to efficiently assume the conformations required for activating the proton transfer. By contrast, the increase in flexibility of the non-stalling mutants compared to wildtype and NC control, allows productive conformations for peptide bond formation to be attained (Supplementary Fig. 11a–c). We also performed additional simulations without the tRNA present in the A-site, where it was observed that the presence of the A-tRNA restricts the movement of the P-site NC (Supplementary Fig. 11d), consistent with the cryo-EM structures when comparing the NC densities in the presence and absence of A-tRNA (Fig. 2 and Supplementary Fig. 5). Taken together, these observations suggest that peptide bond formation is indeed prevented because the Pro-tRNA in the A-site interacts with the P-site nascent chain, together adopting a conformation that is prohibitive for proton transfer in the A-site as well as the subsequent nucleophilic attack onto the P-site peptidyl-tRNA.

## Discussion

In this study, we have determined structures of ribosomes stalled during translation by the RAPP-containing arrest peptides ApdA and ApdP at resolutions of 2.3 and 2.2 Å, respectively. The structures, together with complementary MD simulations and previous biochemistry[16], allow a molecular model to be presented for the mechanism by which the RAPP arrest peptides stall translation by interfering with the process of peptide bond formation. During normal translation, peptide bond formation ensues because the α-amino group of the aminoacyl moiety attached to the A-site tRNA makes a nucleophilic attack on the carbonyl-carbon of the amino acid in the 0 position (the amino acid linked to A76) of the peptidyl-tRNA (Fig. 7a). For the nucleophilic attack to occur, a proton needs to be extracted from the α-amino of the aminoacyl moiety in the A-site, which is thought to be mediated by the 2'O of the ribose of A76 (Fig. 7a). By contrast, during translation of the ApdA and ApdP arrest peptides, the ribosome becomes trapped with the peptidyl-RAP-tRNA in the P-site and the Pro-tRNA in the A-site (Fig. 7b). The PTC of the ribosome is in an induced state with a fully accommodated Pro-tRNA, thus, the stage is set for peptide bond formation to occur, yet, the nucleophilic attack does not take place (Fig. 7b). Our structures and MD simulations indicate that the nucleophilic attack cannot occur because the nitrogen of the Pro in the A-site forms a hydrogen bond with the carbonyl-oxygen of the Ala of the RAP motif in the P-site (Fig. 7b). In this scenario, the 2'O of the ribose of A76 is actually donating, rather than extracting, a proton to the nitrogen of Pro, thereby, also forming a second hydrogen bond, and nullifying any possibility of nucleophilic attack on the peptidyl-tRNA in the P-site (Fig. 7b). Thus, a critical player in our model is the carbonyl-oxygen of the Ala of the RAP motif, which we propose is positioned to interact with the A-site Pro due to structural restraints imposed by the preceding Pro and the following Arg of the RAP motif of the NC (Fig. 7b), explaining why mutations at either Pro or Arg of the RAP motif alleviate stalling[16]. The RAP motifs of ApdA and ApdP do not adopt the β-strand geometry observed for non-stalling peptides where the carbonyl-oxygen of the amino acid in the −1

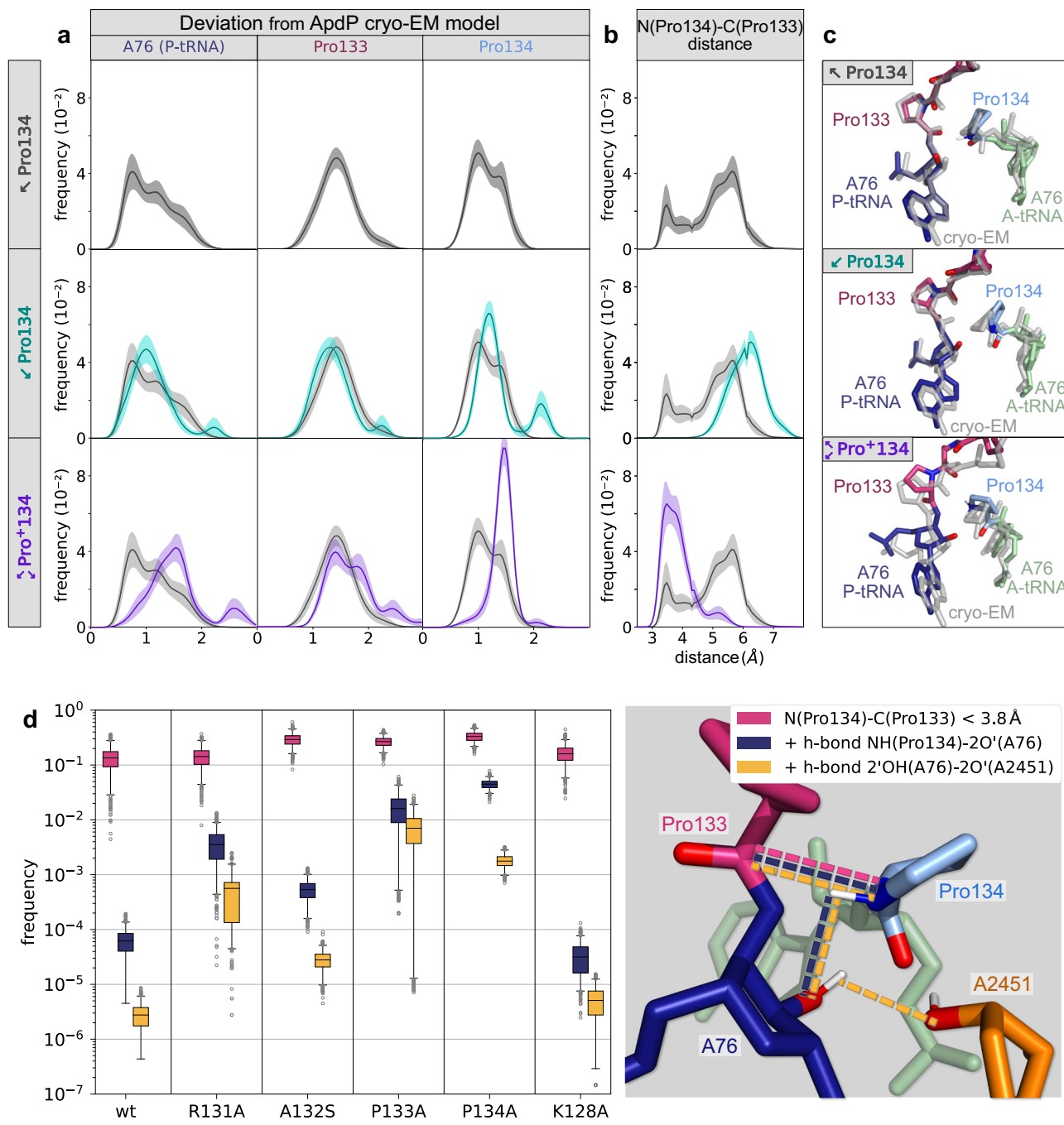

**Fig. 6 | MD simulations of ApdP in the ribosome. a, b** For different protonation states (↖Pro134, ↙Pro134, ⤻Pro⁺134), histograms of deviations (rmsd) from the cryo-EM model (ribose ring of A76, Pro134, and Pro133) (**a**) and of the distances between the α-amino N of Pro134 and the carbonyl C of Pro133 (**b**) are shown. Uncharged protonation states with the N-H pointing either towards the O (Ala132) or towards O2' (A76) are denoted by ↖Pro134 and ↙Pro134, respectively. ⤻Pro⁺134 denotes the charged state with both hydrogens. Lines and error bars in (**a**) and (**b**) were obtained from the mean and standard deviations of 10,000 bootstraps of 20 independent simulations. **c** From the MD simulations of each protonation state, structures corresponding to the most probable rmsd values are shown (colored) and compared with the stalled cryo-EM structure (grey). **d** Frequencies of the conformations fulfilling three conditions required for peptide bond formation. Frequencies of N(Pro134)-C(Pro133) distances lower than 3.8 Å (proximity requirement, magenta). Frequency of conformations which, in addition to the first condition, contain an N-H(Pro134)−2'O(A76) hydrogen bond (blue). Frequency of the conformations that additionally contain the 2'OH(A76) −2'O(A2451) hydrogen bond (yellow). The box plots were obtained by bootstrapping 10,000 samples of 20 independent simulations for each variant. The boxes extend from the first to third quartiles. Whiskers display a 95% confidence interval. Points out of the confidence interval are shown (grey circles). Source data can be obtained from Zenodo (10.5281/zenodo.10426362).

position cannot form a hydrogen bond with the N of the aminoacyl moiety of the A-site tRNA[26,27]. This suggests that the formation of a β-strand geometry by the NC at the PTC may contribute to the efficiency of translation by preventing non-productive NC conformations that interfere with peptide bond formation, as observed here for ApdA and ApdP. We also note that in the absence of the A-site tRNA, we observe that the RAP motif in the ApdA and ApdP NC is relatively flexible (Supplementary Figs. 5, 11), suggesting that the accommodation of the

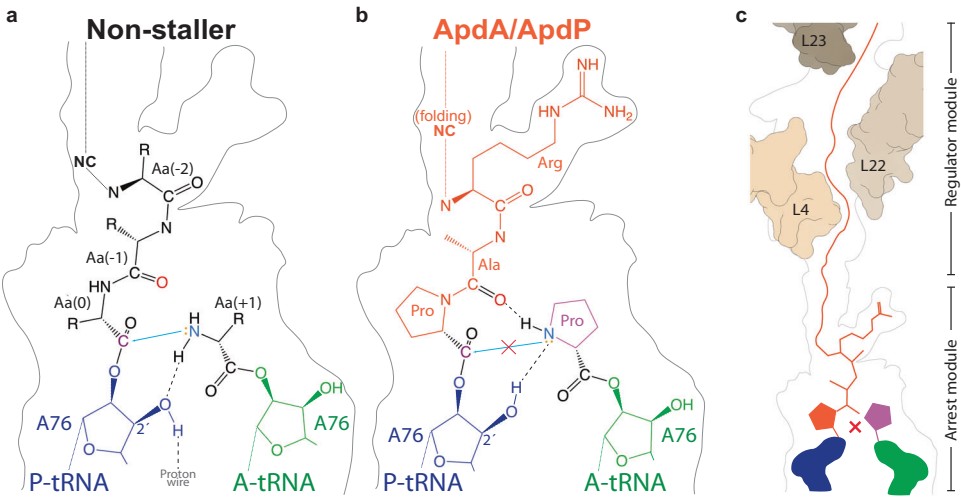

**Fig. 7 | Model for ApdA/ApdP-mediated translational stalling. a, b** Schematic representations of the PTC for **a** canonical non-stalling nascent polypeptide chains, where the lone pair electrons on the α-amino group of the aminoacyl moiety attached to the A-site tRNA makes a nucleophilic attack (blue arrow) on the carbonyl-carbon of the peptidyl-tRNA in the P-site. The nucleophilicity of the α-amino group is increased by the extraction of a proton by the 2′ OH of ribose of A76 of the P-tRNA. **b** RAPP-mediated translation stalling by ApdA or ApdP, where the nucleophilic attack of the nitrogen of the A-site Pro on the carbonyl-carbon of the peptidyl-tRNA cannot occur because (i) the hydrogen of the nitrogen of Pro is involved in a hydrogen bond with the carbonyl-oxygen of Ala of the RAP motif in the P-site, and (ii) the 2′O of the ribose of A76 donates hydrogen to form a hydrogen bond with the lone pair electron, rather than extracting the proton as required for peptide bond formation. **c** ApdA and ApdP stalling is strongly driven by the RAPP arrest module; however, the N-terminal regulator module also contributes by fine-tuning the stalling efficiency.

Pro-tRNA in the A-site contributes to stabilizing the observed conformation. Mutation of the A-site Pro alleviates stalling[16], presumably because, unlike Pro, other amino acids have rotational freedom around the α-amino group, allowing them to more easily adopt optimal geometries for nucleophilic attack and undergo more rapid peptide bond formation (Supplementary Fig. 10).

Despite the similarity of the RAP/P motif present in the ApdA and ApdP with the RAG/P motif present in the SecM, we observe completely different NC conformations and a distinct mechanism of translational inhibition for ApdA and ApdP compared to that reported previously for SecM[20]. Unlike SecM where the Arg of the RAG motif in the P-site is reported to sterically block accommodation of the Pro-tRNA in the A-site[20], we observe that the conformation of the RAP motif in ApdA/ApdP is not only compatible with, but even stabilizes, Pro-tRNA binding at the A-site (Supplementary Fig. 12a–c). These findings were surprising and prompted us to re-determine the structure of a SecM-SRC, where in contrast to ref. 20, we observed that the RAG/P motif of SecM does, in fact, utilize an identical mechanism to stall translation as reported here for ApdA and ApdP[21]. This mechanism is distinct from most other elongation stallers, since they have been shown to generally interfere with the accommodation of the A-site tRNA by promoting an inactive uninduced conformation of the PTC, as exemplified by MifM[22] and VemP[24]. In contrast to VemP, for example, the PTC for ApdA/ApdP is observed in the active induced state with an accommodated A-tRNA (Supplementary Fig. 12d–f). In terms of functional state, the most similar arrest peptide to ApdA/ApdP (with the exception of SecM[21]) is the macrolide-dependent ErmBL arrest peptide, which also traps a pre-attack state of the ribosome with an accommodated A-site tRNA[37]. However, the stalling mechanism differs from ErmBL since peptide bond formation is inhibited because the macrolide induces a rotation of the ErmBL NC that causes displacement of the A76 ribose of the peptidyl-tRNA (Supplementary Fig. 12g–i). Moreover, no interaction is observed between the NC and aminoacyl moiety in the A-site, as observed for ApdA/ApdP (Supplementary Fig. 12g–i).

Despite both containing conserved RAPP motifs, ApdP stalls translation on *B. subtilis* and *E. coli* ribosomes, whereas ApdA stalls translation efficiently only on *B. subtilis* ribosomes[16]. By generating chimeras between ApdA and ApdP, we could localize a region in ApdP that confers this specificity, namely, five residues directly adjacent (N-terminal) to the RAPP motif (Fig. 4). While the conformation of the NC in this region of ApdP adopts a distinct conformation compared to the equivalent region in ApdA (Fig. 3e), the surrounding region of the tunnel encompassing these residues is conserved between *B. subtilis* and *E. coli*. Thus, it remains unclear exactly how these residues contribute to the species-specificity. One possibility is that they are influenced by NC regions deeper in the tunnel where species-specific differences are evident between *B. subtilis* and *E. coli* ribosomes, particularly, the regions involving ribosomal proteins uL4, uL22, and uL23[22]. However, so far, our attempts investigating alterations in these r-proteins, some of which have been documented to alter the species-specificity of MifM stalling[22], were unsuccessful in identifying tunnel components that influence the specificity of ApdA and ApdP stalling (Supplementary Fig. 7).

While the influence of the N-terminal region of the NC on species-specificity of RAPP stalling remains unclear, there is strong evidence that the N-terminal region is important for RAPP stalling. An elegant study by Buskirk and coworkers used a genetic selection for stalling sequences, leading to the identification of RxPP and R/HxPP (amongst others) as strong stalling motifs[38]. Interestingly, when the N-terminal seven residues upstream of RxPP were replaced with the seven residues in front of the HGPP stalling motif (or vice versa), stalling was abolished, illustrating how the N-terminal region can influence, or in this case, alleviate, stalling at these motifs[38]. This study also illustrated that the minimal RxPP motif was sufficient to induce translational arrest, presumably using the mechanism revealed here for ApdA and ApdP, and likely explaining why RxPP motifs are selected against and therefore underrepresented in bacterial proteomes[38]. Collectively, this suggests that RAPP arrest peptides are comprised of two modules, the RAPP "arrest module" that is attached to the P-tRNA and directly involved in preventing peptide bond formation together with a Pro-tRNA in the A-site, as well as a second N-terminal "regulator module" that can modulate the strength (and sometimes even the specificity) of stalling (Fig. 7c).

Recent evidence points to broader implications of the use of the RxPP in bacterial translation regulation beyond proteins involved in the protein localization machinery. For example, in *Bordetella pertussis*, the Gram-negative bacteria responsible for whooping cough, a three gene-operon involved in copper import has been shown to be regulated by an upstream open reading frame containing a conserved RAPP motif[39]. Mutations within the RAPP motif abolished expression of the downstream genes, leading to the suggestion that ribosome stalling occurs at the RAPP motif[39], which we would propose utilizes an analogous mechanism to inhibit peptide bond formation as determined here for ApdA and ApdP. Thus, it seems quite likely that RAPP-like sequences are utilized to regulate the translation of a wide variety of proteins across diverse bacterial phylogenies[40], the full impact of which has yet to be elucidated.

## Methods

### Bacterial strains and plasmids

*B. subtilis strains* (Supplementary Table 1) used for the genetic analysis were constructed by transformation of plasmids (Supplementary Table 2), which were constructed by fusing PCR fragments amplified with PrimeSTAR GXL DNA polymerase (Takara), the template DNAs, and primers (Supplementary Table 3) by Gibson Assembly[41]. Cell extracts for in vitro translation on *B. subtilis* ribosomes were generated as described previously[22] but using the *B. subtilis* strain 168 $\Delta yvyD$ $\Delta ssrA$ $\Delta smpB$ (see below). The protein-coding sequence of ApdA from *Amycolatopsis japonica* and ApdP from *Sinorhizobium medicae*[16] were individually cloned into pDG1662 downstream of a T7 promoter, a ribosome binding site, a His-tag and a Flag-tag using restriction enzymes SphI and HindIII (NEB). The insert of ApdA was amplified by PCR using Q5 High-Fidelity DNA polymerase (NEB) from plasmid pCH2125[16] using primers ApdA-FOR (5´-TACGCTGCATGCGCGGAC-GAGTCGCGCGGGGCGAACGCGACG-3´) and ApdA-REV (5´-AAGCTT TCAGACGGCTACCGGGAAAGGAGGAGCG-3´), whereas the insert of ApdP was amplified by PCR using Q5 High-Fidelty DNA polymerase (NEB) from pCH2126[16] using primers ApdP-FOR (5´-AAAGCATGCAT-CATCGGCCAGAGTGCCGCGTCCCGTG-3´) and ApdP-REV (5´-TTT AAGCTTCGCGAAAGACCTGCCGAACTC-3´). All DNA oligonucleotide primers were purchased from Metabion.

### PCR and in vitro transcription

PCR reaction (Q5 High-Fidelity DNA Polymerase (NEB), Q5 Reaction buffer (NEB) and Q5 High GC Enhancer (NEB)) was used with primers T7FOR (5´- AAATTTTAATACGACTCACTATAGG −3´) and ApdA-REV on the vector harboring the *apdA* gene to generate the amplified DNA sequence (5´- <u>TAATACGACTCACTATAGG</u>GGAATTGTGAGCGGATAAC AATTCCCCACTAGTAATAAT<u>TTTGTTTAACTTTAAGAAGGAGA</u>TATA CC<u>ATG</u>GGCAGCAGCCATCATCATCATCATCAC<u>GATTACAAGGATGAC GACGATAAG</u>GCTAGCAGCAGCGGTACCGGCAGCGGCGAAAACCTCTA TTTTCAGGGTAGTGCGCAAGCATGCGCGGACGAGTCGCGCGGGGCG AACGCGACGGTCGAATCCTCGGTCTCCAAGGCCGTCGCGCCGGTG CGCGCCGCGGGCAGGCTCGCCGCCGAACCCGCGCTCCTGGGTGTGC ACGGTCACGGTGATCTGCCTCTCTTCGGCACCGTCCCGCACGGACC GGCGTCCACGACGCTGCTCCGCCTCAGCGAGCTCCACGACGAAGCA GCTCCCGCCTCGCGCACGGCGAACCGCCTCGCGCTGGGTGATCGCG CTCCTCCTTTCCCGGTAGCCGTC<u>TGA</u>AAGCTT-3´; underlined are the T7 promoter region, ribosomal binding site, start codon, FLAG-tag and stop codon, respectively); primers T7FOR and ApdP-REV were used to perform the same PCR protocol on the vector harboring the *apdP* gene to generate the amplified DNA sequence (5´-<u>TAATACGACTCACTA-TAGG</u>GGAATTGTGAGCGGATAACAATTCCCCACTAGTAATAAT<u>TTTGT TTAACTTTAAGAAGGAGA</u>TATACC<u>ATG</u>GGCAGCAGCCATCATCATCAT CATCAC<u>GATTACAAGGATGACGACGATAAG</u>GCTAGCAGCAGCGGTA CCGGCAGCGGCGAAAACCTCTATTTTCAGGGTAGTGCGCAAGCATGC ATCGGCCAGAGTGCCGCGTCCCGTGCGGGCCGGGGCGCCGGCCGGC AATGTCGCTCAGCCCGATACAGGCTCCTCCGACCGCCCGGTCGCTC

GCCAGATATGCAGGGCGGTTGCGCTGCCCGATCTTCGTTTCATCGG CGAGCGGGCCGATGGCAAGTCATGCTCCGGCGCAGATCCTGCAGCG TTCGTTCCTTTCCAGGCCATTGCCCTGATTGCGCCAGAGATGGCTCC TTCCCTGCCGGTGGCGAAAACCAGAATTGCGCGTCTCCCATCCTGTC AGAGCAAGTGCATTCGCGCGCCGCCAGCGGCGGGAGCCTTCC<u>TTTTGA</u> GAGTTCGGCAGGTCTTTCGCGAAGCTT-3´; underlined are the T7 promoter region, ribosomal binding site, start codon, FLAG-tag and stop codon, respectively). PCR conditions applied were as suggested by the manufacturer and PCR products were purified via spin columns, and in vitro transcription reaction was set up using 1 μg PCR product per 50 μl reaction volume and T7 RNA polymerase (Thermo Fischer Scientific™). RNA was purified by LiCl/ethanol precipitation.

### *Bacillus subtilis* S12 translation extract

The *B. subtilis* S12 translation extract was prepared following a procedure described[22], with some modifications. Briefly, cells (*B. subtilis* strain 168 $\Delta yvyD$ $\Delta ssrA$ $\Delta smpB$) were grown to $OD_{600}$ 0.8 in 2 × YPTG medium (16 g L$^{-1}$ peptone, 10 g L$^{-1}$ yeast extract, 5 g L$^{-1}$ NaCl, 22 mM $NaH_2PO_4$, 40 mM $Na_2HPO_4$, 19.8 g L$^{-1}$ glucose, sterile-filtered) at 37 °C. Cells were collected by centrifugation at 5000 rpm at room temperature for 15 min and subsequently washed 3 × in room temperature Buffer A (10 mM Tris–acetate (pH 8.2, 4 °C), 14 mM magnesium acetate, 60 mM potassium glutamate, 1 mM dithiothreitol, and 6 mM 2-mercaptoethanol, sterile-filtered). After the third wash, cells were resuspended in a minimal volume (0.7 mL g$^{-1}$) of room temperature Buffer B (Buffer A without 2-mercaptoethanol). Cells were snap-frozen in liquid nitrogen and stored at −80 °C. Cells were subsequently thawed on ice and then lysed using FastPrep−24™ MP (4 × 30 min, shaking 4.5 m s$^{-1}$ intercalated by 1 min rest on ice), the lysate was collected by centrifugation (1000 × $g$, 4 °C, 1 min) and further cleared by centrifugation at 12,000 × $g$, 4 °C, 10 min. The lysate was used immediately, or aliquoted, snap-frozen, and stored at −80 °C.

### Generation of stalled-ribosomal complexes

To generate the ApdA-SRC, the ApdA mRNA template (500 ng μL$^{-1}$) was translated by incubation in a *B. subtilis* in vitro translation system. Briefly, a total reaction volume of 450 μL was prepared by mixing 5.85 μL reconstitution buffer, 7.65 μL of methionine, 90 μL amino acid mix, 76.5 μL reaction mix (from RTS 100 HY Kit from Biotechrabbit GmbH) with 90 μL *B. subtilis* S12 translation extract, 135 μL ApdA mRNA, 45 μL 100 mM magnesium acetate, and then incubated for 40 min at 30 °C shaking in a thermomixer (500 rpm).

To generate the ApdP-SRC, the ApdP mRNA template (250 ng μL$^{-1}$) was translated by incubation in a fully reconstituted *E. coli* in vitro PURExpress (NEB) translation system. Briefly, a total reaction volume of 100 μL was prepared by mixing 18 μL water, 40 μL solution A, 12 μL factor mix, 15 μL *E. coli* ribosome (PURExpress® Δ Ribosome Kit, NEB) with 15 μL ApdP mRNA, and then incubated for 40 min at 30 °C shaking in a thermomixer (500 rpm).

### Purification of the stalled-ribosomal complexes

The ApdA-SRC and ApdP-SRC were purified by incubating the respective in vitro translation reactions with 50 μL anti-FLAG® M2 affinity gel (Merck) (previously equilibrated with Hico buffer (50 mM HEPES-KOH (pH 7.4, 4 °C), 100 mM potassium acetate, 15 mM magnesium acetate, 1 mM dithiothreitol, 0.01 % (w/v) *n*-dodecyl-beta-maltoside, sterile-filtered)) inside a Mobicol column fitted with a 35-μm filter (MoBiTec) at 4 °C overnight with rolling. After removal of the flow-through, the beads were washed with a total of 4 mL Hico buffer, and then the bound complex was eventually eluted by incubation with 15 μL Hico buffer containing 0.6 mg mL$^{-1}$ 3xFLAG peptide for 40 min at 4 °C while rolling, followed by centrifugation (2000 × $g$, 4 °C, 2 min). Aliquots from each fraction were checked by Western blotting.

## Preparation of cryo-EM grids and data collection

About 3.5 µl of sample (8 OD$_{260}$/ml) were applied to grids (Quantifoil, Cu, 300 mesh, R3/3 with 3 nm carbon) which had been freshly glow discharged using a GloQube (Quorum Technologies) in negative charge mode at 25 mA for 90 s. Sample vitrification was performed using ethane/propane mix in a Vitrobot Mark IV (Thermo Scientific), with the chamber set to 4 °C and 100% relative humidity, and blotting was performed for 3 s with no drain or wait time. The grids were subsequently mounted into the Autogrid cartridges and loaded onto Talos Arctica (Thermo Fischer Scientific) TEM for screening. Grids were stored in liquid nitrogen until high-resolution data collection. High-resolution data was collected on a Titan Krios microscope aligned for fringe-free imaging and equipped with a Bioquantum K3 (Ametek) direct electron detector. The camera was operated in cor-related double sampling (CDS) mode, and the data were collected at the pixel size of 0.82 Å/px. The microscope condenser system was set to produce 42 e/Å$^2$s electron flux on the specimen, and the data from 1.8 s exposure were stored in 40 frames. The energy-selecting slit was set to 10 eV. The data from 3 × 3 neighboring holes were collected using beam/image shifting while compensating for the additional coma aberration. The data was collected with the nominal defocus range of −0.6 to −1.8 µm. For the ApdA- and ApdP-SRC, a total number of 9930 and 4921 movies were collected, respectively.

## Single-particle reconstruction of SRC complexes

RELION version 3.1[42,43] was used for processing, unless otherwise specified. For motion correction, RELION's implementation of Motion-Cor2 with 4 × 4 patches, and, for initial contrast transfer function (CTF) estimation, CTFFIND version 4.1.14[44], were employed. To estimate local resolution values, Bsoft[45] was used on the half-maps of the final reconstructions (blocres -sampling 0.82 -maxres -box 20 -cutoff 0.143 -verbose 1 -fill 150 -origin 0,0,0 -Mask half_map1 half_map 2) (Supplementary Figs. 2, 4).

**ApdA-SRC dataset.** From 9930 micrographs, 532,749 particles were picked using crYOLO with a general model[46]. In total, 334,479 ribosome-like particles were selected after two-dimensional (2D) classification and extracted at 3× decimated pixel size (2.46 Å per pixel) (Supplementary Fig. 1). An initial three-dimensional (3D) refinement was done using a mol map of an *E. coli* 70S ribosome (PDB ID 7K00 with mRNA and tRNAs removed) as a reference, then partial signal subtraction was performed on the particles to focus on the tRNAs sites, followed by initial 3D classification without angular sampling with six classes. One class containing 70S ribosomes with P-tRNA and substoichiometric A-tRNA (295,256 particles) was subsorted. A class containing 70S with P- and A-tRNAs (142,978 particles) and a class containing 70S with P-tRNA only (152,257 particles) were further processed (Supplementary Fig. 1). In particular, the subtracted particles from the resulting classes were reverted, and 3D and CTF refined (fourth-order aberrations, beam tilt, anisotropic magnification and per-particle defocus value estimation), then subjected to Bayesian polishing[47] and another round of CTF refinement. For the ApdA-SRC with P-tRNA and A-tRNA, a final resolution (gold-standard FSC$_{0.143}$) of masked reconstruction of 2.3 Å was achieved (Supplementary Fig. 2). For the ApdA-SRC with P-tRNA, a final resolution (gold-standard FSC$_{0.143}$) of the masked reconstruction of 2.3 Å was achieved (Supplementary Fig. 5).

**ApdP-SRC dataset.** From 4921 micrographs, 404,941 particles were picked using crYOLO with a general model[46]. In total, 263,503 ribosome-like particles were selected after two-dimensional (2D) classification and extracted at 3× decimated pixel size (2.46 Å per pixel) (Supplementary Fig. 3). An initial three-dimensional (3D) refinement was done using a mol map of an *E. coli* 70S ribosome (PDB ID 7K00 with mRNA and tRNAs removed) as a reference, then partial

signal subtraction was performed on the particles to focus on the tRNAs sites, followed by initial 3D classification without angular sampling with eight classes. One class containing 70S ribosomes with P-tRNA and substoichiometric A-tRNA (205,842 particles) was subsorted into five subclasses, two of which were identical and high-resolution, therefore joined (205,838) for further processing; one class containing 70S with P-tRNA only (52,581 particles) was subsorted into five subclasses, one of which (17,657) was selected for further processing (Supplementary Fig. 3). In particular, the subtracted particles of the resulting classes were reverted and 3D and CTF refined (fourth-order aberrations, beam tilt, anisotropic magnification and per-particle defocus value estimation), then subjected to Bayesian polishing[47] and another round of CTF refinement. For the ApdP-SRC with P-tRNA and A-tRNA, a final resolution (gold-standard FSC$_{0.143}$) of the masked reconstruction of 2.2 Å was achieved (Supplementary Fig. 4). For the ApdP-SRC with P-tRNA, a final resolution (gold-standard FSC$_{0.143}$) of the masked reconstruction of 2.9 Å was achieved (Supplementary Fig. 5).

## Molecular modeling of the SRC complexes

The molecular models of the 30S and 50S ribosomal subunits were based on the *B. subtilis* 70S ribosome (Protein Data Bank (PDB) ID: 6HA1)[48] for ApdA-SRC and on the *E. coli* 70S ribosome (Protein Data Bank (PDB) ID: 5JTE)[37] for ApdP-SRC. The tRNAs and nascent chains were modeled de novo. Restraint files for modified residues were created using aceDRG[49], while the restrain file to link the tRNAs to their aminoacyl-/peptidyl- moiety were kindly provided by Keitaro Yamashita (MRC LMB, UK). The starting models were rigid body fitted using ChimeraX[50] and modeled using Coot 0.9.8.4[51] from the CCP4 software suite version 8.0[52]. The sequence for the tRNAs was adjusted based on the appropriate anticodons corresponding to the mRNA. Final refinements were done in REFMAC 5[53] using Servalcat[54]. The molecular models were validated using Phenix comprehensive cryo-EM validation in Phenix 1.20−4487[55].

## Bacterial in vitro translation arrest assay

In vitro translation arrest assay was carried out using *E. coli*-based coupled transcription-translation system (PUREfrex 1.0; Gene-Frontier), and *Bs* hybrid PURE system[13], in which 1 µM of the *B. subtilis* ribosomes was added instead of *E. coli* ribosome. About 2.5 U/L of T7 RNA polymerase (Takara) was added further to reassure transcription. The DNA templates were prepared by PCR using primers and template DNA listed in Supplementary Table 4. After the translation reaction at 37 °C for 20 min, the reaction was stopped by adding three volumes of 1.3 x SDS-PAGE loading buffer (167 mM Tris-HCl (pH 6.8), 2.7% (wt/vol) SDS, 20% (vol/vol) glycerol, 6.7 mM DTT, a trace amount of bromophenol blue), and, when indicated, samples were further treated with 0.2 mg/ml RNase A (Promega) at 37 °C for 10 min to degrade the tRNA moiety of peptidyl-tRNA immediately before electrophoresis.

## Eukaryotic in vitro translation arrest assay

The DNA templates were prepared by PCR using primers and templates listed in Supplementary Table 4. In vitro transcription was carried out using T7 RNA Polymerase ver.2.0 (TaKaRa) and 175 ng of PCR product per 10 µl reaction volume. The mRNA was then purified by RNAClean XP (Beckman Coulter) and used for in vitro translation using the Rabbit Reticulocyte Lysate (RRL) translation system (Promega). A total reaction volume of 4 µL was prepared by mixing 2.8 µL Rabbit Reticulocyte Lysate (Nuclease-Treated), 10 µM Amino Acid Mixture Minus Methionine, and 10 µM Amino Acid Mixture Minus Leucine with the 40 ng/µL mRNA. After the translation reaction at 30 °C for 20 min, the reaction was stopped by adding seven volumes of 1.1 x SDS-PAGE loading buffer (143 mM Tris-HCl (pH 6.8), 2.3% (wt/vol) SDS, 17% (vol/vol) glycerol, 5.7 mM DTT, a trace amount of bromophenol blue), and, when indicated, samples were further treated with 0.2 mg/ml RNase A

(Promega) at 37 °C for 10 min to degrade the tRNA moiety of peptidyl-tRNA immediately before electrophoresis.

## Western blotting

Samples were separated by 10% polyacrylamide gel prepared with WIDE RANGE Gel buffer (Nacalai Tasque), transferred onto a PVDF membrane, and then subjected to immuno-detection using primary antibodies against GFP (mFX75, 012−22541; Wako) or FLAG-tag (M2, F3165; Sigma) at 1/3000 and 1/5000 dilutions, respectively, and the secondary antibody against mouse IgG-HRP (170-6516; Bio-Rad) at 1/5000 dilution. The Images were obtained and analyzed using Amersham Imager 600 (GE Healthcare) luminoimager.

## β-galactosidase assay

*B. subtilis* cells were cultured in LB medium at 37 °C and withdrawn at an optical density at 600 nm ($OD_{600}$) of 0.5–1.0 for β-galactosidase assay. About 100 μL portions of the cultures were transferred to individual wells of a 96-well plate, and $OD_{600}$ was recorded. Cells were then lysed with 50 μL of Y-PER reagent (Thermo Scientific) for 20 min at room temperature. After 30 μL of o-nitrophenyl-β-D-galactopyranoside (ONPG) in Z-buffer (60 mM $Na_2HPO_4$, 40 mM $NaH_2PO_4$, 10 mM KCl, 1 mM $MgSO_4$, 38 mM β-mercaptoethanol) was added to each well, $OD_{420}$ and $OD_{550}$ were measured every 5 min over 60 min at 28 °C. Arbitrary units [AU] of β-galactosidase activity were calculated by the formula $[(1000 \times V_{420} - 1.3 \times V_{550})/OD_{600}]$, where $V_{420}$ and $V_{550}$ are the first-order rate constants, $OD_{420}$/min and $OD_{550}$/min, respectively.

## Molecular dynamics (MD) simulations

From the model of the ApdP-SRC with both A- and P-site tRNAs, we extracted all residues within 35 Å of the ApdP NC and used it as a starting structure for the MD simulations of the wild-type simulation system. In order to test which protonation state of Pro134 is the most compatible with the cryo-EM model, we considered three possible scenarios: ↖Pro134, where Pro134 is uncharged and the α-amino hydrogen points towards the O (Ala132), (ii) ↙Pro134, where Pro134 is uncharged and the α-amino hydrogen is oriented towards the O2′ (A76), and (iii) ⌄Pro⁺134, where the proline is charged and the two α-amino hydrogens point towards the O (Ala132) the O2′ (A76) respectively. The distinction between the ↖Pro134 and the ↙Pro134 states was required since classical MD simulations do not allow nitrogen pyramidal inversion to be observed and, hence, interconversion between the two scenarios.

To obtain the dynamics of ApdP NC variants shown to alleviate stalling[16], we also performed MD simulations with the variants: P134A, P133A, A132S, and R131A. As a negative control, we also included the K128A variant, which does not affect the stalling efficiency[16]. The starting structures for the ApdP variants were obtained by mutating single residues in the wild-type model with the mutagenesis tool of PyMOL (Schrödinger). Since we observed that the structural ensemble with the ↖Pro134 protonation state agreed best with the cryo-EM structure (Fig. 6a), we used this protonation state for the P133A, A132S, R131A, and K128A variants. Ala134 in the P134A variant was modeled as uncharged.

The protonation states of the histidine residues were determined using WHAT IF[56]. Water molecules, $K^+$, and $Mg^{2+}$ ions from the cryo-EM structure were included in the starting structure. Each structure was positioned at the center of a dodecahedral box, with a minimum distance of 1.5 nm between the atoms and the box boundaries (Supplementary Table 5). The system was then solvated using the program solvate[57] and neutralized by using the program GENION to replace water molecules with $K^+$ ions[57]. GENION was additionally used to add 7 mM $MgCl_2$ and 150 mM KCl. All simulations were performed with GROMACS 2018[57] using the amber14sb forcefield[58], the OPC water model[59], and the microMg parameters from ref. 60 for the $Mg^{2+}$ ions,

and the $K^+$ and $Cl^-$ parameters from ref. 61. Partial charges of N-terminal residues with a neutral α-amino group are not included by default in the Amber force field. Therefore, we computed the charges of Pro134 and Ala134 in their uncharged states with GaussView 5.0.8[62], using *N*-methylamide as the capping group for the C-terminus of the two amino acids. Geometry optimization of the molecules and charge calculation were both carried out with the Hartree-Fock method[63], using the 6-31 G basis set[64]. To obtain charges compatible with the Amber force field, the Restrained Electrostatic Potential (RESP) method was applied to fit the charges obtained from the Hartree-Fock calculation[65]. During the RESP fitting, the partial charges of the *N*-methylamide group were restrained to the ones already provided in the force field. Lennard–Jones and short-range electrostatic interactions were calculated within a distance of 1 nm. For distances beyond 1 nm, long-range electrostatic interactions were computed by particle-mesh Ewald summation[66] with a 0.12-nm grid spacing. Bond lengths were constrained using the LINCS algorithm[67]. Temperature coupling to a heat bath at $T = 300$ K was performed independently for solute and solvent using velocity rescaling[68] with a coupling time constant of $\tau_T = 0.1$ ps. Virtual site constraints[69] were applied for hydrogen atoms allowing for an integration time step of 4 fs. Coordinates were recorded every 5 ps.

Initially, energy minimization by steepest descent was performed on each system by applying harmonic position restraints ($k = 1000$ kJ mol⁻¹ nm⁻¹) to the solute-heavy atoms. After that, for each minimized system, 20 simulations were carried out, each one consisting of two equilibration steps and one production run. In the first equilibration step (0−50 ns), the pressure was coupled to a Berendsen barostat Berendsen, 1984 #17353} ($\tau_p = 1$ ps) and position restraints ($k = 1000$ kJ mol⁻¹ nm⁻¹) were applied on all the heavy atoms of the solute. In the second equilibration step (50−70 ns), the position restraints were linearly decreased to zero for all the heavy atoms of the solute positioned within 25 Å from the P-site peptide. The force constant of the restraining potential applied to the heavy atoms placed in the outer shell (25–35 Å) was decreased to the one obtained from the fluctuations previously observed in full-ribosome simulations[70]. Finally, during the production run (70−2070 ns), the Parrinello-Rahman barostat[71] ($\tau_p = 1$ ps) was used, keeping the position restraints only on the outer-shell heavy atoms. The 20 simulations performed for each system sum up to a total simulation time of 40 μs of production run per system and 360 μs in total. The trajectories were analyzed using GROMACS[57], Python 3.8.5 (https://www.python.org), and Pandas 1.1.3 (https://pandas.pydata.org/). The results were plotted using Matplotlib 3.3.2 (https://matplotlib.org/). The first 200 ns of each production run were excluded from the analyses to allow equilibration of the system.

## Structural deviations

To identify the protonation state of Pro134 that is most compatible with the cryo-EM data of the ribosome in complex with ApdP, we quantified, for each protonation state (↖Pro134, ↙Pro134, and ⌄Pro⁺134), the deviation of the simulated conformational ensemble from the cryo-EM structure. To that aim, the root mean square deviation (rmsd) of the residues and bases directly involved in peptide bond formation (P-site tRNA A76, Pro133, and Pro134) from their cryo-EM conformation was calculated. The rmsd values were computed for each frame after the rigid-body fitting of all the P-atoms of the system. Histograms of the rmsd values were then obtained using 80 bins (Fig. 6a). To identify the two most dominant conformational modes of Ala132, we performed principal component analysis (PCA) of the Ala132 backbone atoms positions. First, all trajectories of wildtype (↖Pro134 protonation state) and variants were superimposed by least-square fitting of the positions of the 23S rRNA phosphate atoms. Then the trajectories were concatenated and the atomic displacement

covariance matrix was calculated. Finally, for each system, the trajectories of each replica were projected on the first two eigenvectors of this matrix. Histograms of the projection values were then obtained using 80 bins for each conformational mode (Supplementary Fig. 11c). To obtain statistical uncertainties, 10,000 (for the rmsd values) or 1000 (for the PCA) combinations of 20 replicas were randomly selected for each simulated system and the analysis was repeated on each subset. The mean and standard deviation of all subsets were then computed.

### Monitoring distances relevant to peptide bond formation

The distances between the α-amino N of Pro134 and the carbonyl C of Pro133 were calculated for each frame of the ⭦Pro134, ⭧Pro134, and ⭦Pro⁺134 simulations. The distributions of the N(Pro134)-C(Pro133) distances were used to further evaluate which of the three systems is most similar to the cryo-EM structure (Fig. 6a). To obtain statistical uncertainties, 10,000 combinations of 20 replicas were randomly selected for each state, and the analysis was repeated on each subset. The mean and standard deviation of all subsets were then computed.

To identify the mechanism of ApdP stalling, we monitored distances that are relevant to fulfill the conditions required for peptide bond formation, i.e., the proximity between the α-amino N and the C involved in the peptide bond, and deprotonation of the α-amino group. For identifying conformations that satisfy the first condition, we monitored the N(Pro134)-C(Pro133) distances. To assess conformations meeting the second condition, we computed the distances between the N-H of Pro134 and the 2'O of A76, since a hydrogen bond between these atoms is necessary for the α-amino nitrogen to lose one hydrogen. Additionally, we identified conformations compatible with the proton wire mechanism of deprotonation[26] by monitoring the distances between the 2'OH of A76 and 2'O of A2451. In order to compare how frequently stalling and non-stalling variants accessed productive conformations, we counted the number of frames where the proximity and the α-amino group deprotonation were fulfilled. The counts were then divided by the total number of simulation frames. We considered the proximity condition fulfilled for conformations with N(Pro134)-C(Pro133) distances lower than 3.8 Å. To account for hydrogen bonds solely between N-H of Pro134 and 2'O of A76, while excluding bifurcated hydrogen bonds with O(Ala132), we identified conformations as productive when the N-H(Pro134)−2'O(A76) distance was <3 Å and the N-H(Pro134)-O(Ala132) distance >3 Å. We considered the 2'OH(A76)−2'O(A2451) hydrogen bond formed when the distance between the two atoms was lower than 3 Å.

### Structural flexibility

In order to assess the structural flexibility of the ApdP peptide variants and of the peptide in the absence of the A-site tRNA, the root mean square fluctuation (rmsf) was computed for the backbone of each residue of the peptide after aligning the MD trajectory frames using the P-atoms of the simulated region of the 23S rRNA.

### Figures

UCSF ChimeraX 1.3 was used to isolate density and visualize density images and structural superpositions. Hydrogen bonds were determined using the default settings of ChimeraX 1.3, (distance between donor and acceptor <3.4 Å with an angle of 120° ± 20°). Models were aligned using PyMol version 2.4 (Schrödinger). Figures were assembled with Adobe Illustrator (the latest development release, regularly updated).

### Reporting summary

Further information on research design is available in the Nature Portfolio Reporting Summary linked to this article.

### Data availability

Micrographs have been deposited as uncorrected frames in the Electron Microscopy Public Image Archive (EMPIAR) with the accession codes EMPIAR-11698 (ApdP-SRC) and EMPIAR-11702 (ApdA-SRC). Cryo-EM maps have been deposited in the Electron Microscopy Data Bank (EMDB) with accession codes EMD-18332 (*ApdA-SRC with A- and P-site tRNA*), EMD-18341 (*ApdA-SRC with P-site tRNA only*), EMD-18320 (*ApdP-SRC with A- and P-site tRNA*), EMD-18340 *ApdP-SRC with P-site tRNA only*). Molecular models have been deposited in the Protein Data Bank with accession codes 8QCQ (*ApdA-SRC with A- and P-site tRNA*) and 8QBT (*ApdP-SRC with A- and P-site tRNA*). Publicly available data used included PDB ID 1VY4, 8CVK, 6OLG, 3JBU, 5NWY, and 5JTE. Initial coordinates, input files and output coordinates of the MD simulations, residue topologies of the uncharged terminal Alanine and Proline (used for modeling Ala134 and Pro134), rmsd, distances, rmsf values obtained from the MD trajectories, and projections on the most dominant conformational modes sampled by the Ala132 backbone atoms are publicly available on Zenodo (10.5281/zenodo.10426362). Source data are provided with this paper.

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

## Acknowledgements

We thank Satoshi Naito and Tomoya Imamichi for providing a plasmid for XBP1u, Machiko Murata and Naoko Muraki for their technical support. This work was supported by the Deutsche Forschungsgemeinschaft (DFG) (grant WI3285/11-1 to D.N.W.), JSPS Grant-in-Aid for Scientific Research (Grant No. 16H04788, 26116008, 20H05926, and 21K06053 to S.C and 19K16044 and 21K15020 to K.F), Institute for Fermentation, Osaka (grant G-2021-2-063 to S.C.), and under Germany's Excellence Strategy grant no. EXC 2067/1-390729940 (L.V.B.). We acknowledge financial support from the Open Access Publication Fund of Universität Hamburg. Cryo-EM grid preparation was performed at the Multi-User Cryo-EM Facility at the Centre for Structural Systems Biology, Hamburg, supported by the Universität Hamburg and DFG grant numbers (INST 152/772–1 | 152/774–1 | 152/775–1 | 152/776–1 | 152/777–1 FUGG). We would like to thank Jiří Nováček for data collection via iNEXT-Discovery project number 23828 (funded by the Horizon 2020 program of the European Commission) and acknowledge the cryo-electron microscopy and tomography core facility CEITEC MU of CIISB, Instruct-CZ Centre supported by MEYS CR (LM2018127).

## Author contributions

M.M. generated samples and processed the microscopy data, as well as generated and refined the molecular models and made all the structure figures. B.B. generated strains and helped with sample preparation. H.P. prepared and screened the cryo-EM grids and helped with processing and refinement. K.F. and S.C. performed the biochemical stalling assays. S.G. and L.V.B. performed the MD simulations. D.N.W. and M.M. wrote the manuscript with input from all authors. D.N.W. and S.C. conceived and supervised the project.

## Funding

## Competing interests

The authors declare no competing interests.
