## [Peer Review File · Nature Communications]

RAPP-containing arrest peptides induce translational stalling by short circuiting the ribosomal peptidyltransferase activityREVIEWER COMMENTS

Reviewer #1 (Remarks to the Author):

In this paper, Morici and co-workers investigate how ribosomes are stalled by the “RAPP” arrest peptide which has been found in a number of bacteria and regulate the expression of key proteins. In this study, the authors have used cryo-electron microscopy to solve structures of *B.subtilis* and *E.coli* ribosomes arrested with two novel peptides, ApdA and ApdP; ApdA. The AdpP can stall both *B.subtilis* and *E.coli* ribosomes, while AdpA only stalls *B.subtilis* ribosomes. The authors reveal that the RxPP motif prevents peptide bond formation from taking place because the proline occupying the A-site adopts an unfavourable conformation. The authors demonstrate this finding using a combination of cryoEM and MD simulations. The use of MD simulations to support the author’s analysis of the protonation state of the proline residue was a fine effort. Such an analysis was made possible due to the excellent quality of the maps that were obtained following a rigorous processing procedure.

The combined use of cryoEM and biochemical analysis reveals clearly the stalling properties of the RxPP motif and a mechanism of action. The curiosity also lies in why the two peptides, ApdA and ApdP, each with the RxPP motif can stall ribosomes from different species uniquely, where only the ApdP stalls both *E.coli* and *B.subtilis*. The authors show using chimeras (Fig. 4) that the minimum stalling sequence from AdpA and AdpD sequence for *B.subtilis*, is RxPP. which explains why there is an effect on changes in stalling when the uL4, uL22 or uL23 proteins were modified (Supplementary Figure 7). By contrast, the minimum sequence for stalling for in *E.coli*, appears to be QSKCIRAPP, indicating that uL4 and uL22 may be involved. In the case of species-selectivity, evaluating from the perspective of *E.coli*, where the sequence requirement for stalling appears to be more stringent (QSKCIRAPP), might offer more information (particularly if more distal regions of the ribosome are involved in stalling, e.g., ribosomal proteins in the tunnel or around the PTC, see point (2) below). It is possible that understanding inherent differences in translational arrest across species may fall beyond the scope of this study, however any additional experimental insights that the authors may have on this point would be welcome.

Overall, this comparative study is meticulous, the choice of experiments sound and their execution is to a very high standard. The paper is of great importance for the field and sheds light into mechanisms translational regulation and how this occurs across organisms. In addition, this study has a range of broader implications, not least of these, is using such information to develop novel antibiotic targets. Below are some minor considerations that would benefit from clarification by the authors:

1. The cryoEM maps of ApdA- and ApdP-(Δ A-tRNA) are comparable in resolution to those of ApdA and ApdP (2.3 and 2.9Å). The ApdA-(Δ A-tRNA) is especially similar to the ApdA and ApdP reconstructions in terms of resolution and where the particle numbers differ by almost 10K compared to the ApdA. Supplementary Figure 5 shows nice base separation for both structures lacking A-tRNA, and the Pro122

and Pro133 have better resolved density compared to the ApdP-stalled nascent chain. Can the authors comment further on the sigma level of the maps in Fig 2 and Supplementary Figure 5?

2. As it is pertinent to their analysis of the A and P sites with the PTC, can the authors comment on the local resolution for ApdA- and ApdP-(Δ A-tRNA) obtained around the PTC? Similarly, are there any mRNA-tRNA pairs that with visible bases? Is there any risk that tRNA heterogeneity may exist which may skew the current analysis?

3. Often, bL27 has a long unstructured N-terminal extension, but which is visible in several bacterial structures. (e.g. E.coli and N. gonorrhoeae at comparable resolution) with an occupied A-site. Is the bL27 extension observed in the author's maps, and if so, could they comment on whether bL27 could also be involved in stalling by RAPP?

4. Proline can have up to 8 rotamers with 3 of them are more prevalent than the rest. Can the authors clarify how they determined the correct proline rotamer, as it is not clear from the maps in Fig 2.

Reviewer #2 (Remarks to the Author):

The manuscript by Morici et al. reports structures of bacterial ribosomes stalled on RAPP peptide motifs. Ribosome stalling by iterated proline residues has been well characterized over the past 10+ years, but the mechanism of stalling has not been clearly established. As the only secondary amine-containing amino acid, it has been fascinating to rationalize why an A-site Pro-tRNA imposes constraints during the peptidyl transferase reaction. Here, the authors present compelling structural evidence that explains the mechanisms underlying this phenomenon. The structural work is excellent and comprehensive. I have only a few very minor suggestions that may improve the presentation of the work.

1. Can the authors comment on the variable occupancies of A-site tRNA between the ApdA and ApdP cryo-EM datasets? Specifically, is there anything to be gleaned from ApdA particles containing similar ratios 1 or 2-bound tRNAs, whereas the ApdP particles contain mostly 2-bound tRNAs?

2. It would be helpful to include some labels for the tRNA bases in Figure 2b and 2e (e.g., A76).

3. It is not clear what the differences are between Figs. 2a and 2c, as well as between 2d and 2f. The figure captions for 2c and 2f indicate that they are views of the 3D-refined ribosomes, but how is that different from 2a and 2d?

4. The criteria for the potential hydrogen bonds shown in Fig. 3 should be explained (i.e., distance cutoff?).

5. The arrows in Fig. 5g and h are hard to see and should be enlarged.

Reviewer #3 (Remarks to the Author):

In the manuscript by Morici et al., the authors describe the molecular mechanism by which a common class of stalling peptides found in bacteria arrests translation. These peptides have a conserved RAPP motif, which seems similar to the well-described stalling motif in the SecM stalling peptide. However, as the authors show using a combination of cryo-EM, biochemistry, cell-based experiments, and MD simulations, the RAP sequence motif stalls translation by preventing peptide bond formation from a fully-accommodated Pro-tRNA substrate in the ribosomal A site. The authors report cryo-EM structures at 2.2-2.3 Å resolution, which show the subtle reorientation of the peptidyl-tRNA and positioning of the proline on the Pro-tRNA in the A site. The attacking nucleophile on proline is positioned further from the carbonyl carbon than expected, forming an H-bond to the carbonyl from the alanine in the RAP motif. The authors argue that this H-bond and the singly-protonated nature of the attacking nucleophile switches the role of the 2'-O on A76 of the P-site substrate from an H-bond acceptor to donor. This switch would prevent the 2'-O from serving its role in nucleophile deprotonation that is thought to be part of the normal peptidyl transferase reaction. Overall, the paper is well written and the range of experiments support the model presented by the authors. However, there are some points of confusion that the authors could clarify.

1. It's striking that all 3 waters in the proton shuttle mechanism seem to be present (Supplementary Fig. 9), but the authors don't present this clearly in the manuscript. At least one of these panels should be in the main text. To make room, panels a-c of Fig. 5 could be moved to the supplementary figures, as the structure with fMet in the P site is not as relevant to the present story compared to the peptidyl tRNA in Syroegin et al. 2023, and is likely not the same quality due to some flexibility in the P site for such a short substrate. The authors show beautiful density for waters further down the exit tunnel in Fig. 3, but these are arguably not as important to present as waters 1-3 in the putative PTC reaction mechanism! The authors could make a figure similar to ED Fig. 4 in Syroegin et al. for the main text.

2. The authors should also spend a bit more time discussing the rotation of the Ala carbonyl relative to what's seen in a peptide not terminated with Pro (Fig. 4d-f). Why does this occur? Would it be the case for any amino acid preceding a Pro at the C-terminus of the peptidyl-tRNA? Does the Arg force the issue? Where are the backbone angles on Ramachandran plots for the structures presented here and in Syroegin et al.?

3. Panel 4g should be replaced with the analogous figure using Fig. 4d coordinates, since a peptidyl-tRNA substrate is more relevant to the present manuscript.

4. For the MD simulations, what happens to waters 1-3 in the PTC? Are they essentially invariant? This is an important part of the mechanism that should be included in Fig. 6d.

5. Also for the MD simulations, what happens to the carbonyl on the Ala (or mutated residue at this position)? Does it stay in place? This could also be an important part of Fig. 6, to better understand the mechanism of stalling.

Reviewer #4 (Remarks to the Author):

The manuscript "Novel arrest peptides induce ribosome stalling by short circuiting the ribosomal peptidyltransferase activity" by Moricy et al, describes the mechanism by which the RAPP arrest motif 'stalls' translation.

To do this the authors solved the structures of the stalled ribosomes by cryo EM and performed extensive MD simulations to elucidate the mechanism atomistically.

The manuscript is clear, well written and interesting. And deserves publication.

Maybe in the results section, some more details on the MD simulations should be reported.

REVIEWER COMMENTS

We thank all four reviewers for their positive comments and valuable feedback that we believe has improved the manuscript. We hope we have satisfactorily addressed all the comments in the point-by-point response below.

We have also made a number of additional revisions that change one of the overall conclusions of the manuscript, namely, that we now think that the stalling mechanism of RAPP motifs present in ApdA and ApdP is in fact analogous to that of the RAGP motif present in SecM. The reason for this change in interpretation is that we have in the meantime re-determined a structure of a ribosome stalled during translation of the full-length SecM arrest peptide at 2.0 Å resolution. These findings are included as a separate manuscript (also submitted to Nat Comm) that we have appended as a related manuscript to this resubmission. In this manuscript, we clearly show that the mechanism of RAGP-mediated stalling by SecM is analogous to that of RAPP-mediated stalling of ApdA and ApdP. For this reason, we have reworked the ApdA/ApdP manuscript to make it clear that our results differ from that of the previous reported structure of SecM at 3.0-3.7 Å from Zhang et al 2015, and that we believe that this structure is simply an artefact (for many reasons outlined in the new SecM manuscript) and does not represent the actual mechanism of SecM-mediated stalling. We are very curious to see whether the reviewers also agree with our analysis and conclusions in this respect. Text changes to this effect are to be found in the final paragraph of the introduction on page 4, lines 89-94 as well as in the discussion on page 14, lines 426-429.

Reviewer #1 (Remarks to the Author):

In this paper, Morici and co-workers investigate how ribosomes are stalled by the “RAPP” arrest peptide which has been found in a number of bacteria and regulate the expression of key proteins. In this study, the authors have used cryo-electron microscopy to solve structures of B.subtilis and E.coli ribosomes arrested with two novel peptides, ApdA and ApdP; ApdA. The AdpP can stall both B.subtilis and E.coli ribosomes, while AdpA only stalls B.subtilis ribosomes. The authors reveal that the RxPP motif prevents peptide bond formation from taking place because the proline occupying the A-site adopts an unfavourable conformation. The authors demonstrate this finding using a combination of cryoEM and MD simulations. The use of MD simulations to support the author’s analysis of the protonation state of the proline residue was a fine effort. Such an analysis was made possible due to the excellent quality of the maps that were obtained following a rigorous processing procedure.

The combined use of cryoEM and biochemical analysis reveals clearly the stalling properties of the RxPP motif and a mechanism of action. The curiosity also lies in why the two peptides, ApdA and ApdP, each with the RxPP motif can stall ribosomes from different species uniquely, where only the ApdP stalls both E.coli and B.subtilis. The authors show using chimeras (Fig. 4) that the minimum stalling sequence from AdpA and AdpD sequence for B.subtilis, is RxPP. which explains why there is an effect on changes in stalling when the uL4, uL22 or uL23 proteins were modified (Supplementary Figure 7). By contrast, the minimum sequence for stalling for in E.coli, appears to be QSKCIRAPP, indicating that uL4 and uL22 may be involved. In the case of species-selectivity, evaluating from the perspective of E.coli, where the sequence requirement for stalling appears to be more stringent (QSKCIRAPP), might offer more information (particularly if more distal regions of the ribosome are involved in stalling, e.g., ribosomal proteins in the tunnel or around the PTC, see point (2) below). It is possible

that understanding inherent differences in translational arrest across species may fall beyond the scope of this study, however any additional experimental insights that the authors may have on this point would be welcome.

We also thought that it was interesting that ApdP and ApdA exhibit different species-specificity, and this observation was the original reason we started the project to determine structures of both arrest peptides. However, given the flexibility of the NC in the lower regions of the tunnel, we were not able to provide structural insight into the specificity. We agree that one could examine whether mutations in uL4, uL22 and uL23 in E. coli ribosomes can influence the stalling of ApdP, as we tried for B. subtilis, however, we have not undertaken such experiments. We are reluctant to perform such experiments since even if we were to observe an effect on ApdP stalling in one of the variants, we cannot relate this to the structure due to the flexibility of the NC within this region, and therefore such experiments will not enable to unravel the structural basis for the specificity.

Overall, this comparative study is meticulous, the choice of experiments sound and their execution is to a very high standard. The paper is of great importance for the field and sheds light into mechanisms translational regulation and how this occurs across organisms. In addition, this study has a range of broader implications, not least of these, is using such information to develop novel antibiotic targets. Below are some minor considerations that would benefit from clarification by the authors:

1. The cryoEM maps of ApdA- and ApdP-(Δ A-tRNA) are comparable in resolution to those of ApdA and ApdP (2.3 and 2.9Å). The ApdA-(Δ A-tRNA) is especially similar to the ApdA and ApdP reconstructions in terms of resolution and where the particle numbers differ by almost 10K compared to the ApdA. Supplementary Figure 5 shows nice base separation for both structures lacking A-tRNA, and the Pro122 and Pro133 have better resolved density compared to the ApdP-stalled nascent chain. Can the authors comment further on the sigma level of the maps in Fig 2 and Supplementary Figure 5?

The reviewer is correct that the quality of the ApdA-SRC without A-tRNA is similar in terms of average resolution to the ApdA- and ApdP-SRCs that contain A-tRNA. However, while this is also reflected in the quality of the rRNA and tRNA where one can indeed see base separation for the ApdA-SRC without A-tRNA, the quality of the density for the nascent chain, including Pro122 of ApdA and Pro133 of ApdP, is very poor. The density is not defined and cannot be modelled de novo, unlike in the A-site containing maps (Fig. 2) where even the hole in the Pro122 and Pro133 can be observed. The NC models used in Sup Fig 5 are simply aligned from the respective structures containing A-tRNA. This has nothing to do with the sigma levels of the maps - the NCs are simply more flexible in the absence of A-tRNA, which is also supported by our MD simulations (see Sup Fig. 11 for example). Nevertheless, the approximate sigma levels used for ApdP and ApdA are 2.5σ and 2.2σ in Figure 2, and 1.3σ and 1.5σ for Sup Fig. 5 i.e. the sigma levels are higher for the Fig 2 images than the Sup Fig 5 images.

2. As it is pertinent to their analysis of the A and P sites with the PTC, can the authors comment on the local resolution for ApdA- and ApdP-(Δ A-tRNA) obtained around the PTC? Similarly, are there any mRNA-tRNA pairs that with visible bases? Is there any risk that tRNA heterogeneity may exist which may skew the current analysis?

The local resolution of the ApdA- and ApdP-(ΔA -tRNA) obtained around the PTC is very good, enabling unambiguous modeling of the rRNA nucleotides. The density for the mRNA-tRNA pairs is also consistent with the assignment of the P-tRNA as bearing Pro122 for the ApdA-SRC. By contrast, the worse resolution of the ApdP-SRC means that we cannot exclude the possibility of tRNA heterogeneity. But based on available biochemistry (Sakiyama et al 2021), we have no reason to believe that there should be another stalling site.

3. Often, bL27 has a long unstructured N-terminal extension, but which is visible in several bacterial structures. (e.g. E.coli and N. gonorrhoeae at comparable resolution) with an occupied A-site. Is the bL27 extension observed in the author's maps, and if so, could they comment on whether bL27 could also be involved in stalling by RAPP?

The N-terminus is indeed ordered in the A-tRNA containing maps and adopts an identical conformation observed in the pre-attack state structures. This is seen in the Sup Fig. 9. Note that the very N-terminal residue for E. coli is Ala2 but is Ala10 for B. subtilis (the N-terminus of B. subtilis undergoes cleavage). As the reviewer most likely knows, the N-terminus of bL27 becomes ordered upon accommodation of the A-site tRNA, as seen also in the ApdA- and ApdP-SRCs. We don't see why bL27 should be involved in the RAPP-mediated stalling mechanism since our model is that the extraction of the proton from the N of the Pro moiety of the A-site tRNA by the 2'O of A76 of the P-site tRNA cannot take place, i.e. an event that occurs well before any proposed involvement of bL27 in the proton wire.

4. Proline can have up to 8 rotamers with 3 of them are more prevalent than the rest. Can the authors clarify how they determined the correct proline rotamer, as it is not clear from the maps in Fig 2.

During modelling we considered all 8 proline rotamers, three of which are the most prevalent, as the reviewer correctly pointed out. One of the three most prevalent rotamers, namely the C exo rotamer, fitted the density best and maintained its conformation during refinement, therefore, this rotamer was used in our models. However, we make no claims about which proline rotamer is present and, in our opinion, to do so would require even higher local resolution than we have here. Its also possible that the Pro adopts multiple rotamers and we observe only an average, however, higher resolution would be required to address such issues and its not clear that this is relevant for the mechanism.

Reviewer #2 (Remarks to the Author):

The manuscript by Morici et al. reports structures of bacterial ribosomes stalled on RAPP peptide motifs. Ribosome stalling by iterated proline residues has been well characterized over the past 10+ years, but the mechanism of stalling has not been clearly established. As the only secondary amine-containing amino acid, it has been fascinating to rationalize why an A-site Pro-tRNA imposes constraints during the peptidyl transferase reaction. Here, the authors present compelling structural evidence that explains the mechanisms underlying this phenomenon. The structural work is excellent and comprehensive. I have only a few very minor suggestions that may improve the presentation of the work.

1. Can the authors comment on the variable occupancies of A-site tRNA between the ApdA and ApdP cryo-EM datasets? Specifically, is there anything to be gleaned from ApdA particles containing similar ratios 1 or 2-bound tRNAs, whereas the ApdP particles contain mostly 2-bound tRNAs?

*It's a good question, but unfortunately, it's hard to know whether the different ratios of P vs A+P substates between the ApdA and ApdP datasets is due to the differences in the arrest peptides themselves (ApdA vs ApdP), the different ribosomal species used (*B. subtilis* vs *E. coli*) or even the different in vitro translation assays used (*E. coli* PURExpress system vs homemade lysate-based system for *B. subtilis*). In fact, we have no idea whether this is technical or biological issue, hence, we prefer not to make any claims in relation to this point.*

2. It would be helpful to include some labels for the tRNA bases in Figure 2b and 2e (e.g., A76).

We have now included labels for the tRNA nucleotides in Figure 2b and 2e as requested

3. It is not clear what the differences are between Figs. 2a and 2c, as well as between 2d and 2f. The figure captions for 2c and 2f indicate that they are views of the 3D-refined ribosomes, but how is that different from 2a and 2d?

Panels 2a and 2d are the post-processed maps, not the 3D-refined maps. This is mentioned in the legend.

4. The criteria for the potential hydrogen bonds shown in Fig. 3 should be explained (i.e., distance cutoff?).

For all analyses, we use the default cut-off of ChimeraX which considers a hydrogen-bond if the potential donor and acceptor are separated by $<3.4 \text{ \AA}$ with an angle of $120^\circ \pm 20^\circ$. We have now added this to the methods on page 26, line 804-805.

5. The arrows in Fig. 5g and h are hard to see and should be enlarged.

We have enlarged the arrows in Fig 5g and 5h as requested.

Reviewer #3 (Remarks to the Author):

In the manuscript by Morici et al., the authors describe the molecular mechanism by which a common class of stalling peptides found in bacteria arrests translation. These peptides have a conserved RAPP motif, which seems similar to the well-described

stalling motif in the SecM stalling peptide. However, as the authors show using a combination of cryo-EM, biochemistry, cell-based experiments, and MD simulations, the RAP sequence motif stalls translation by preventing peptide bond formation from a fully-accommodated Pro-tRNA substrate in the ribosomal A site. The authors report cryo-EM structures at 2.2-2.3 Å resolution, which show the subtle reorientation of the peptidyl-tRNA and positioning of the proline on the Pro-tRNA in the A site. The attacking nucleophile on proline is positioned further from the carbonyl carbon than expected, forming an H-bond to the carbonyl from the alanine in the RAP motif. The authors argue that this H-bond and the singly-protonated nature of the attacking nucleophile switches the role of the 2'-O on A76 of the P-site substrate from an H-bond acceptor to donor. This switch would prevent the 2'-O from serving its role in nucleophile deprotonation that is thought to be part of the normal peptidyl transferase reaction. Overall, the paper is well written and the range of experiments support the model presented by the authors. However, there are some points of confusion that the authors could clarify.

1. It's striking that all 3 waters in the proton shuttle mechanism seem to be present (Supplementary Fig. 9), but the authors don't present this clearly in the manuscript. At least one of these panels should be in the main text. To make room, panels a-c of Fig. 5 could be moved to the supplementary figures, as the structure with fMet in the P site is not as relevant to the present story compared to the peptidyl tRNA in Syroegin et al. 2023, and is likely not the same quality due to some flexibility in the P site for such a short substrate. The authors show beautiful density for waters further down the exit tunnel in Fig. 3, but these are arguably not as important to present as waters 1-3 in the putative PTC reaction mechanism! The authors could make a figure similar to ED Fig. 4 in Syroegin et al. for the main text.

We agree with the reviewer that the structure of the pre-attack state with fMet-tRNA in the P-site depicted in panels 5a-c is not as relevant as those in d-f, but in our opinion, the pre-attack structure in a-c is a more common reference for the pre-attack state, which is why we prefer to include it. With respect to the 3 waters in the proton shuttle, we do not consider them important for the mechanism of the arrest peptides, therefore, we would rather not include them in the main Figure 5. The reason is, as for bL27 response to point 3 of reviewer #1, the involvement of the waters in the proton shuttle/wire occurs after the proton is extracted from N of the Pro moiety in the A-site. For this reason, we prefer to leave these images in the Supplementary Information showing that the PTC is primed for peptide-bond formation. Nevertheless, we have now reproduced the images in Sup Fig 9 in the style similar to ED Fig. 4 in Syroegin et al., as requested by the reviewer.

2. The authors should also spend a bit more time discussing the rotation of the Ala carbonyl relative to what's seen in a peptide not terminated with Pro (Fig. 4d-f). Why does this occur? Would it be the case for any amino acid preceding a Pro at the C-terminus of the peptidyl-tRNA? Does the Arg force the issue? Where are the backbone angles on Ramachandran plots for the structures presented here and in Syroegin et al.?

We agree with the reviewer that we should have mentioned that the carbonyl-oxygen in the non-stalling scenario, such as the pre-attack state observed in Fig 5, is not within hydrogen bonding distance/geometry. This was an oversight and we have now added text in the discussion on page 14 (lines 407-409) relating to this. We also thank the reviewer for the point about the Ramachandran plots suggesting that non-stalling peptides adopts beta-strand

conformations. This is not the case for ApdA or ApdP (see image below) and we now mention the implications of this in the same section on page 14, lines 409-412.

In the discussion, we mention that we think that the unusual positioning of the carbonyl-oxygen of Ala to form a hydrogen bond with the Pro moiety in the A-site is likely to arise “due to structural restraints imposed by the preceding Pro and the following Arg of the RAP motif of NC (Fig. 7b), explaining why mutations at either Pro or Arg of the RAP motif alleviate stalling (Sakiyama et al., 2021).” We would predict that the alanine is not critical for stalling since in some organisms, alanine is substituted to glycine in ApdP and mutation of Ala to Ser had a less dramatic effect on the stalling efficiency (Sakiyama et al 2021), however, we cannot rule out that other mutations would abolish stalling. Moreover, Buskirk and coworkers (Woolstenhulme et al 2013) also identified a variety of RxPP motifs that stall translation – this is mentioned in the discussion on page 16. Buskirk and coworkers also demonstrated that stalling at RAPP motifs could be suppressed by the N-terminal region of the nascent chains, presumably by preventing the RAPP motif adopting the arrest peptide conformation observed in our structures, i.e. preventing the carbonyl-oxygen of Ala forming a hydrogen bond with the Pro moiety on the A-site tRNA. We have now included this point in the discussion (page 15, lines 467-471) and propose that such RAPP arrest peptides are composed of two modules, an arrest module (RAPP) and an N-terminal regulator module that can suppress or maybe in some cases even enhance translational stalling. We have added a new panel c into Figure 7 to illustrate this point.

3. Panel 4g should be replaced with the analogous figure using Fig. 4d coordinates, since a peptidyl-tRNA substrate is more relevant to the present manuscript.

We presume the reviewer is referring to Figure 5. We have replaced the panel 5g with the pre-attack state coordinates that were used to generate Figure 5d.

4. For the MD simulations, what happens to waters 1-3 in the PTC? Are they essentially invariant? This is an important part of the mechanism that should be included in Fig. 6d.

As mentioned above in relation to point 1, we do not think it is an important part of the mechanism and therefore have not included them in Figure 6d. This notion is supported by our observation that the waters are invariant in the MD simulations of all stalling and non-stalling ApdP variants. Here, we show the fluctuations (rmsf) of the waters in the MD simulations and their distances from the cryo-EM positions. Right panel: cryo-EM positions in red, MD positions in yellow. The radius of each sphere indicates the fluctuations (rmsf).

5. Also for the MD simulations, what happens to the carbonyl on the Ala (or mutated residue at this position)? Does it stay in place? This could also be an important part of Fig. 6, to better understand the mechanism of stalling.

In the simulations of the non-stalling ApdP variants which include a mutation of the peptide (R131A, A132S, and P133A), the Ala132 (or Ser132 in the A132S variant) samples conformations further away from the cryo-EM position than for the wt ApdP or the control (K128A). This is also reflected in the increased fluctuations of this residue (Supplementary Figure 11a). To show the direction in which the residue moves, we performed a principal component analysis. The movement along the two most dominant conformational modes is now shown Supplementary Figure 11c and referred to on page 13, line 372.

Reviewer #4 (Remarks to the Author):

The manuscript "Novel arrest peptides induce ribosome stalling by short circuiting the ribosomal peptidyltransferase activity" by Moricy et al, describes the mechanism by which the RAPP arrest motif 'stalls' translation.

To do this the authors solved the structures of the stalled ribosomes by cryo EM and performed extensive MD simulations to elucidate the mechanism atomistically.

The manuscript is clear, well written and interesting. And deserves publication.

Maybe in the results section, some more details on the MD simulations should be reported.

We have now added a few more details on the MD simulations in the results section as requested. This was anyway necessary to address point 5 of reviewer #3.

REVIEWERS' COMMENTS

Reviewer #1 (Remarks to the Author):

The authors have addressed all the points raised and the revised version will be a great addition to the ribosome field.

Reviewer #2 (Remarks to the Author):

The revised manuscript by Morici et al. addressed my previous suggestions and has been further improved by the complementary manuscript (Gersteur*, Morici* et al.). These manuscripts present a compelling mechanism for ribosome stalling by the R-A-P/G-P motif. Specifically, their structural analysis reveals that the stalling motifs stabilize a state in which the geometry of the A-site proline prevents peptide bond formation at the peptidyltransferase center.

For the sake of consistency between the two papers, it may be helpful for the authors to comment on elements that make up the N-terminal "regulator module" in the ApdA/ApdP structures. Are there elements within the exit tunnel that are similarly ordered as observed in their SecM structures? This question is pertinent as it is a major punchline in the SecM stalling mechanism and a feature of their ApdA/ApdP model figure (Fig. 7c).

Overall, this manuscript strengthens and is strengthened by its companion manuscript, and I recommend publication.

Reviewer #3 (Remarks to the Author):

In this revised version of the manuscript, the authors have addressed the critiques raised in the previous round of review, and strengthened the paper. As the SecM paper is running slightly "behind" this one in the review process, I won't comment specifically on the SecM-related changes to this manuscript except to say the SecM study is clearly a separate story that would be interesting to see back-to-back with this one.

Reviewer #4 (Remarks to the Author):

The authors have addressed all my concerns.